# Measuring the Impact of Equal Treatment as Blindness via Distributions of Explanations Disparities

## Abstract

Liberal political philosophy advocates for the policy of *equal treatment as blindness*, which seeks to achieve fairness by treating individuals without considering their protected characteristics directly. However, this policy has faced longstanding criticism for perpetuating existing inequalities. In machine learning, this policy can be translated into the concept of *fairness as unawareness*, and be measured using disparate treatment metrics such as Demographic Parity (a.k.a. Statistical Parity). Our analysis reveals that Demographic Parity does not faithfully measure whether individuals are being treated independently of the protected attribute by the model. We introduce the Explanation Disparity metric to measure fairness under *equal treatment as blindness* policy. Our metric evaluates the fairness of predictive models by analyzing the extent to which the protected attribute can be inferred from the distribution of explanation values, specifically using Shapley values. The proposed metric tests for statistical independence of the explanation distributions over populations with different protected characteristics. We show the theoretical properties of "Explanation Disparity" and devise an equal treatment inspector based on the AUC of a Classifier Two-Sample Test. We experiment with synthetic and natural data to demonstrate and compare the notion with related ones. We release `explanationspace`, an open-source Python package with methods and tutorials.

## 1 Introduction

Liberal-oriented politics, put forward by scholars such as Friedman et al. (1990) and Nozick (1974), often advocate for a policy of *equal treatment as blindness*, where fairness is aimed to be achieved by making decisions without directly considering protected attributes. This principle can be translated to machine learning as fairness by unawareness, where protected attributes are excluded from the training data (Simons et al., 2021; Cornacchia et al., 2023; Chen et al., 2019a; Fabris et al., 2023). However, this policy of ignoring the protected attribute has been historically widely criticized, by gender and ethnic studies (MacKinnon, 1989; Crenshaw, 1997; Minow, 1990), for example:

> *A notion of equality that demands disregarding a "difference" calls for assimilation to an unstated norm. To strip away difference, then, is often to remove or ignore a feature distinguishing an individual from a presumed norm [. . . ] but leaving that norm in place as the measure for equal treatment.*
> `Martha Minow, 1990.  Making all the difference.  59–60`

These critiques suggest that simply ignoring the protected attribute does not achieve equality; instead, it risks perpetuating the very inequalities it seeks to eliminate. By analyzing the college admission use case in Appendix A, we collect a set of requirements towards measuring the criticism of this policy and argue that existing fairness metrics do not manage to account for inequalities in this policy. For example, *individual fairness* (Dwork et al., 2012) cannot identify group discrimination. The measure of *equal opportunity* (Hardt et al., 2016) focuses on discrepancies in error rates rather than differences in treatment and necessitates labelled data. Similarly, *causal fairness* (Plecko and Bareinboim, 2024) demands background knowledge. Unfortunately, both labelled data and sufficient background knowledge are often unavailable in many practical

scenarios. Demographic Parity (DP), also called Statistical Parity, which compares the distributions of predicted outcomes of a model $f$ for different social groups has been proposed as a potential metric to measure this effect by researchers(Simons et al., 2021; Heidari et al., 2019; Wachter et al., 2020). However, we show that Demographic Parity may indicate fairness, even when groups are being treated differently by the model.

We propose the fairness metric of Explanation Disparity for supervised machine learning models, to measure the longstanding criticism of the policy of *equal treatment as blindness*. The notion of Explanation Disparity considers the contribution of non-protected features to the machine learning model output as explained by Shapley values (Lundberg and Lee, 2017). If two social groups are treated the same, the distributions of feature contributions, which we call *explanation distributions*, will not be distinguishable. Thus, Explanation Disparity tests the independence of Shapley values with the protected feature, via a classifier two-sample test that infers the protected attribute. We introduce a decision tool, the "Equal Treatment Inspector", that implements this idea. When detecting unequal treatment, it explains the features involved in such inequality, supporting understanding the roots of unequal treatment in the machine learning model.

In summary, our contributions are:

1. The definition of Explanation Disparity metric, that measures the independence of the protected attribute based on explanation distributions.
2. The definition of an "Equal Treatment Inspector" workflow, based on a classifier two-sample test, for recognizing and explaining un-equal treatment.
3. The study of the formal relationships between Demographic Parity and Explanation Disparity.
4. Extensive experiments, both on synthetic and natural datasets, to demonstrate our method and compare it with related work.
5. An open-source Python package `explanationspace` implementing the "Equal Treatment Inspector", which is `scikit-learn` compatible, and includes documentation and tutorials.

## 2 Foundations and Related work

Table 1: Fairness metrics alignment with criticism of *equal treatment as blindness* criteria. Requirements are derived from a use case explained in Appendix A, and metrics are further discussed and defined in Section A.3

| Metric | Group Discrimination (R1) | Unlabelled Data (R2) | No Background Knowledge (R3) | Proxy Discrim. (R4) | Explanation Capabilities (R5) |
|---|---|---|---|---|---|
| Explanation Disparity | ✓ | ✓ | ✓ | ✓ | ✓ |
| Demographic Parity | ✓ | ✓ | ✓ | ✗ | ✗ |
| Conditional DP | ✓ | ✗ | ✓ | ✗ | ✗ |
| Equal Opportunity | ✓ | ✗ | ✓ | ✗ | ✗ |
| Treatment Equality | ✓ | ✗ | ✓ | ✗ | ✗ |
| Feature Importance Disp. | ✓ | ✓ | ✓ | ✗ | ✓ |
| Counterfactual Fairness | ✓ | ✓ | ✗ | ✓ | ✓ |
| Fairness-Unawareness | ✓ | ✓ | ✓ | ✗ | ✗ |
| Individual Fairness | ✗ | ✓ | ✓ | ✓ | ✓ |

We briefly survey the philosophical and technical foundations of our contribution as well as related work. We build on Shapley values, which are generally known in the machine learning community, but we provide their mathematical definition in Appendix B for the paper to be self-contained.

### 2.1 Basic Notation and Definitions of Fairness

A supervised learning model is a function $f_\theta : \mathbf{X} \to Y$ induced from a set of observations, called the training set, where $\mathbf{X} = \{X_1, \ldots, X_p\}$ are $p$ predictive features, $Y$ is the target feature, and $\theta$ are the models' parameters. The domain of the target feature is $dom(Y) = \{0, 1\}$ (binary classification) or $dom(Y) = \mathbb{R}$ (regression). For

binary classification, we assume a probabilistic classifier, and we denote by $f_\theta(\mathbf{x})$ the estimate of the probability $P(Y = 1|\mathbf{X} = \mathbf{x})$ over the (unknown) distribution of $\mathbf{X} \times Y$. For regression, $f_\theta(\mathbf{x})$ estimates $E[Y|\mathbf{X} = \mathbf{x}]$. A dataset of observations from the distribution of $\mathbf{X} \times Y$ is denoted by $\mathcal{D} = \{(\mathbf{x}_1, y_1), \dots, (\mathbf{x}_n, y_n)\} \sim \mathbf{X} \times Y$. We call the projection of $\mathcal{D}$ on $\mathbf{X}$, written $\mathcal{D}_\mathbf{X} = \{\mathbf{x}_1, \dots, \mathbf{x}_n\} \sim \mathbf{X}$, the empirical *input distribution*. The dataset $f_\theta(\mathcal{D}_\mathbf{X}) = \{f_\theta(\mathbf{x}) \mid \mathbf{x} \in \mathcal{D}_\mathbf{X}\}$ is called the empirical *prediction distribution*.

We assume a feature $Z$ representing social groups, called the *protected feature*, and assume it to be binary valued in the theoretical analysis – thus modeling membership to a protected social group. $Z$ can either be included in the predictive features $\mathbf{X}$ or not. If not, we assume that it is still available for a test dataset.

We write $\mathbf{A} \perp \mathbf{B}$ to denote statistical independence between the two sets of random variables $\mathbf{A}$ and $\mathbf{B}$, or equivalently, between two multivariate probability distributions. We define two common fairness notions and corresponding fairness metrics that quantify a model's degree of discrimination or unfairness (Mehrabi et al., 2022).

**Definition 2.1.** *(Demographic Parity (DP))*. A model $f_\theta$ achieves Demographic Parity if $f_\theta(\mathbf{X}) \perp Z$.

Thus, Demographic Parity holds if $\forall z. P(f_\theta(\mathbf{X})|Z = z) = P(f_\theta(\mathbf{X}))$. For binary $Z$'s, we can derive an unfairness metric as $d(P(f_\theta(\mathbf{X})|Z = 1), P(f_\theta(\mathbf{X})))$, where $d(\cdot)$ is a distance between probability distributions.

**Definition 2.2.** *(Equal Opportunity (EO))* A model $f_\theta$ achieves equal opportunity if $\forall z. P(f_\theta(\mathbf{X})|Y = 1, Z = z) = P(f_\theta(\mathbf{X}) = 1|Y = 1)$.

Unfairness can be measured for binary $Z$'s as $d(P(f_\theta(\mathbf{X})|Y = 1, Z = 1), P(f_\theta(\mathbf{X}) = 1|Y = 1))$.

## 2.2 Philosophical Foundations and Computable Fairness Metrics

Political and moral philosophers from the **egalitarian** school of thought often consider *equal opportunity* to be the key promoter of fairness and social justice, providing qualified individuals with equal chances to succeed regardless of their background or circumstances Rawls (1958; 1991); Dworkin (1981a;b); Arneson (1989); Cohen (1989). In fair-ML, Hardt et al. (2016) proposed translating *equal opportunity* into the inter-group difference of the true positive rates. Heidari et al. (2019) provided a moral framework to ground such a metric of Equal Opportunity. Gabriel (2022) explored the relationship between AI and principles of Rawlsian distributive justice. Kuppler et al. (2021); Baumann et al. (2022) discuss the mismatches of egalitarian distributive justice with fairness metrics. From a machine learning perspective, the technical drawback is that metrics for *equal opportunity* require label annotations for true positive outcomes, which are not always available after the deployment of a model. Acquiring labelled data post-deployment poses a considerable challenge and is often impractical, or even impossible, exhibiting well-known biases, such as confounding, selection, and missingness (Feng et al., 2023; Ruggieri et al., 2023).

The **liberalism** school of thought argues that individuals should be treated equally independently of outcomes (Friedman et al., 1990; Nozick, 1974). *Equal treatment* has also been defined as "equal treatment-as-blindness" or neutrality Sunstein (1992); Miller and Howell (1959). A policy that can be translated as "fairness through unawareness" (Grgic-Hlaca et al., 2018; Cornacchia et al., 2023; Chen et al., 2019a; Fabris et al., 2023), which considers an algorithm fair if the protected attribute(s) $Z$ is not explicitly used in decision-making. Any mapping $f_\theta : \mathbf{X} \to Y$ that excludes $Z$ from $\mathbf{X}$ satisfies this policy. The criticism relies that even without learning from or using the protected feature, a model can discriminate against the protected groups through correlated features in $\mathbf{X}$ as a proxy of the protected one (Pedreschi et al., 2008).

From a technical perspective, a potential metric to measure the impact of *equal treatment as blindness* policy can be Demographic Parity or Statistical Parity (used synonymously) (Dwork et al., 2012). As we will analyze in Section 4.1, Demographic Parity implies that demographic groups experience the same distribution of model outcomes. However, we might still observe Demographic Parity while individuals from different groups are treated differently. One instance is when the contribution of some features that correlates with the protected attributes cancel out (Ruggieri et al., 2023). In a hypothetical hiring scenario, this might represent a situation in which men are hired for a reason (e.g., they have university degrees), while women are hired for a different reason (e.g. they have industrial working experience). In this situation, Demographic Parity might still be satisfied, but individuals of different groups are treated differently, thus not accurately measuring the

criticism of *equal treatment as blindenss* political philosophy policy. Our metric of Explanation Disparity remedies this drawback.

Demographic Parity implies that two demographic groups experience the same distribution of model outcomes, even if the first has much better prospects for achieving the predicted outcome. Thus, a model $f$ that achieves Demographic Parity may have to prefer individuals from one group over those from another, violating the requirement for equal treatment of all individuals. In Appendix A.3, we illustrate the use-case of college admissions, where equal treatment is desirable, but neither equal opportunity nor equal outcomes achieve it.

### 2.3 Related Work

We briefly review related works below. See Appendix C for an in-depth comparison.

**Related Fairness Metrics**  In recent years, fairness metrics have proliferated. Yet, a significant gap exists in their alignment with distributive justice principles and political philosophy (Baumann et al., 2022; Hertweck et al., 2024; Fazelpour and Lipton, 2020; Fazelpour et al., 2022). To bridge this gap in the criticism of liberal-oriented politics, our focus is on establishing a set of criteria that align with these principles. *(R1) Group Discrimination*, decisions based on protected characteristics that individuals did not choose at birth. *(R2) Unlabelled Data* metrics should prioritize assessing disparities in treatment rather than fixating on error disparities. *(R3) No Background Knowledge* not necessitating understanding of causal or structural aspects. *(R4) Mitigating Proxy Discrimination* detecting that certain features may indirectly contribute to biased decisions. *(R5) Explanation Capability* offer both theoretical underpinnings and empirical validation of the sources driving discrimination. We add a base requirement, *(R0) Predictive Performance*, which requires the model $f_\theta$ to have predictive performance, as a random classifier achieves several fairness metrics. In Table 1 and Appendix A.3, we present a comparison of various fairness metrics against these criteria. Furthermore, in the detailed use case presented in Appendix A, we extract and emphasize philosophical requirements to underscore the alignment with liberal political philosophy.

**Testing for Demographic Parity (DP)**  Most related works measure DP using statistics such as Mann–Whitney, Kolmogorov-Smirnov or Wasserstein distance, and related statistical tests (Raji et al., 2020; Kearns et al., 2018; Cho et al., 2020). Other research lines have aimed at measuring DP when the protected attribute is a continuous variable (Jiang et al., 2022). We also measure and test for DP through the AUC of a classifier two-sample test. Detailed comparisons are reported in Appendix G.5.

**Classifier Two-Sample Test (C2ST)**  We formalize a classifier two-sample test (C2ST) based on the AUC to measure the independence of sets of random variables. Lopez-Paz and Oquab (2017) explored C2ST using accuracy metrics. Chakravarti et al. (2023) used AUC for C2ST without providing a formal proof of correctness. Moreover, we use a Brunner-Munzel test statistics, which exhibits a better power than the previous two approaches – more in Appendix G.1.

**Explainability for Fair Supervised Learning**  Lundberg (2020), and the related work by Chang et al. (2023), apply Shapley values to DP. Their approach can be rephrased as decomposing $f_\theta(\mathbf{X}) \perp Z$ by examining $\mathcal{S}(f_\theta, \mathbf{X})_i \perp Z$ individually for every Shapley value component $\mathcal{S}(f_\theta, \mathbf{X})_i$, with $i \in [1, p]$. First, they intend to tackle DP, not the Explanation Disparity notion we will introduce. Moreover, even when considering DP, their approach suffers from statistical pitfalls as we show in Appendix A.3.2. Other recent lines of work assume knowledge about causal relationships between random variables, such as Grabowicz et al. (2022). Our work does not rely on knowledge of causal graphs but exploits the Shapley values' theoretical properties to obtain fairness model auditing guarantees.

## 3 A Model for Monitoring Equal Treatment

### 3.1 Formalizing Equal Treatment

To establish a criterion for equal treatment, we rely on the notion of explanation distributions.

**Definition 3.1.** *(Explanation Distribution)* An explanation function $\mathcal{S} : \mathcal{F} \times \mathbf{X} \to \mathbb{R}^p$ maps a model $f_\theta \in \mathcal{F}$ and an input instance $\mathbf{x} \in \mathbf{X}$ into a vector of reals $\mathcal{S}(f_\theta, \mathbf{x}) \in \mathbb{R}^p$. An (empirical) *explanation distribution* is then defined on an input distribution $\mathcal{D}_\mathbf{X}$ as $\mathcal{S}(f_\theta, \mathcal{D}_\mathbf{X}) = \{\mathcal{S}(f_\theta, \mathbf{x}) \mid \mathbf{x} \in \mathcal{D}_\mathbf{X}\} \subseteq \mathbb{R}^p$.

We use Shapley values as an explanation function (cf. Appendix B). In Appendix H, we discuss the usage of LIME. Next, we introduce our new fairness criterion, Explanation Disparity, which asks for the independence of a model's explanations from the protected feature.

**Definition 3.2.** *(Explanation Disparity (ED))* A model $f_\theta$ achieves ED if $\mathcal{S}(f_\theta, \mathbf{X}) \perp Z$.

Such a definition aims to characterize the philosophical notion of *equal treatment* by encoding the "treatment" performed by the model through the attribution of the importance of its input features. As we will see later in Section 4, Explanation Disparity is a stronger metric than Demographic Parity since it not only requires that the distributions of the predictions are similar but that the processes of how predictions are made (i.e., the explanations) are also similar.

## 3.2 Equal Treatment Inspector

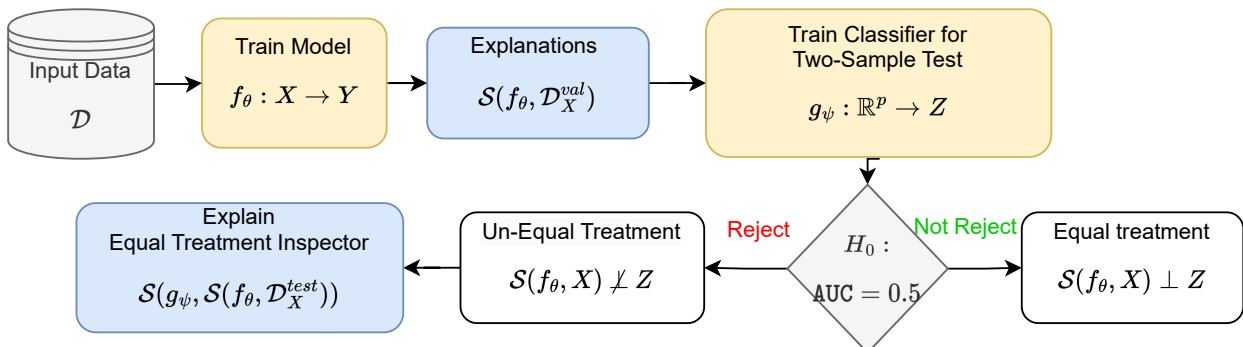

Figure 1: Equal Treatment Inspector workflow. The model $f_\theta$ is learned based on training data, $\mathcal{D}^{tr} = \{(\mathbf{x}_i, y_i)\}$, and outputs the explanations $\mathcal{S}(f_\theta, \mathcal{D}_\mathbf{X}^{val})$. The C2ST receives the explanations to predict the protected attribute, $Z$ on validation data $\mathcal{D}^{val}$. The AUC of the C2ST classifier $g_\psi$ on test data $\mathcal{D}^{te}$ decides for or against *equal treatment*. We can interpret the driver for unequal treatment on $g_\psi$ with explainable AI techniques.

Our approach is based on the properties of the Shapley values (cf. Appendix B) and on a novel classifier two-sample test. We partition the available data into three sets $\mathcal{D}^{tr}, \mathcal{D}^{val}, \mathcal{D}^{te} \subseteq \mathbf{X} \times Y$. $\mathcal{D}^{tr}$ is the training set of $f_\theta \in \mathcal{F}$ (not required if $f_\theta$ is already trained). The dataset $\{(\mathcal{S}(f_\theta, \mathbf{x}), z_\mathbf{x}) \mid \mathbf{x} \in \mathcal{D}_{\mathbf{X} \setminus Z}^{val}\}$, where $z_\mathbf{x} \in Z$ is the social group of $\mathbf{x}$, is used to train the "Equal Treatment Inspector", $g_\psi$. Here, $g_\psi$ is any ML method with parameters $\psi$ that optimizes a loss function $\ell$:

$$\psi = \arg\min_{\tilde{\psi}} \sum_{\mathbf{x} \in \mathcal{D}_{\mathbf{X} \setminus Z}^{val}} \ell(g_{\tilde{\psi}}(\mathcal{S}(f_\theta, \mathbf{x})), z_\mathbf{x}) \tag{1}$$

To evaluate whether there is an equal treatment violation, we perform a statistical test of independence based on the AUC of $g_\psi$ on a test set $\mathcal{D}^{te}$ (and related social group, if not already included in it). We also use $\mathcal{D}^{te}$ for testing the approach w.r.t. baselines. Besides detecting fairness violations, a common desideratum is to understand which specific features drive such violations. The "Equal Treatment Inspector" $g_\psi$ can provide information on *the features' contribution to the un-equal treatment* either by-design, if it is an interpretable model, or by applying post-hoc explanations techniques, e.g., by applying again Shapley values on $g_\psi$. See Figure 1 for a visualization of the whole workflow.

## 4 Theoretical Analysis

### 4.1 Explanation Disparity vs Demographic Parity vs Fairness of the Input

We start by observing that Explanation Disparity (independence of the explanation distribution from the protected attribute) is a sufficient condition for Demographic Parity (independence of the prediction distribution from the protected attribute).

**Lemma 4.1.** *If $\mathcal{S}(f_\theta, \mathbf{X}) \perp Z$ then $f_\theta(\mathbf{X}) \perp Z$.*

*Proof.* By the propagation of independence in probability distributions, the premise implies $(\sum_i \mathcal{S}_i(f_\theta, \mathbf{X}) + c) \perp Z$ where $c$ is any constant. By setting $c = E[f(\mathbf{X})]$ and by the efficiency property (6), we have the conclusion. □

Therefore, a DP violation (on the prediction distribution) is also a ED violation (in the explanation distribution). ED accounts for a stricter notion of fairness. In general, the other direction does not hold (in Appendix D, we study a simple condition when ED and DP are equivalent). We can have dependence of $Z$ from the explanation features, but the sum of such features cancels out resulting in perfect DP on the prediction distribution. This issue is also known as the Yule's effect (Ruggieri et al., 2023).

**Example 4.1.** Consider the model $f(\mathbf{x}) = \mathbf{x}_1 + \mathbf{x}_2$. Let $Z \sim Ber(0.5)$, $A \sim U(-3, -1)$, and $B \sim N(2, 1)$ be independent, and let us define:

$$\mathbf{X}_1 = A \cdot Z + B \cdot (1 - Z) \quad \mathbf{X}_2 = B \cdot Z + A \cdot (1 - Z)$$

We have $f(\mathbf{X}) = A + B \perp Z$ since $A, B, Z$ are independent. Let us calculate $\mathcal{S}(f, \mathbf{X})$ in the two cases $Z = 0$ and $Z = 1$. If $Z = 0$, we have $\mathbf{X}_1 = B, \mathbf{X}_2 = A$, and then $\mathcal{S}(f, \mathbf{X})_1 = B - E[B] = B - 2 \sim N(0, 1)$ and $\mathcal{S}(f, \mathbf{X})_2 = A - E[A] = A + 2 \sim U(-1, 1)$. Similarly, for $Z = 1$, we have $\mathbf{X}_1 = A, \mathbf{X}_2 = B$, and then $\mathcal{S}(f, \mathbf{X})_1 = A - E[A] = A + 2 \sim U(-1, 1)$ and $\mathcal{S}(f, \mathbf{X})_2 = B - E[B] = B - 2 \sim N(0, 1)$. This shows:

$$P(\mathcal{S}(f, \mathbf{X})|Z = 0) \neq P(\mathcal{S}(f, \mathbf{X})|Z = 1)$$

and then $\mathcal{S}(f, \mathbf{X}) \not\perp Z$. Notice this example holds both for the interventional and the observational variants of Shapley values, as we consider a linear model over independent features (cf. Lemma B.1).

Statistical independence between the input $\mathbf{X}$ and the protected attribute $Z$, i.e., $\mathbf{X} \perp Z$, is another fairness notion. It targets fairness of the (input) dataset, disregarding the model $f_\theta$. For fairness-aware training algorithms, which are able not to (directly or indirectly) rely on $Z$, violation of such a notion of fairness does not imply ED violation nor DP violation.

**Example 4.2.** Let $\mathbf{X}$ be three independent features such that $E[\mathbf{X}_1] = E[\mathbf{X}_2] = E[\mathbf{X}_3] = 0$, and $\mathbf{X}_1, \mathbf{X}_2 \perp Z$, and $\mathbf{X}_3 \not\perp Z$. The target feature is defined as $Y = \mathbf{X}_1 + \mathbf{X}_2$, hence it is also independent from $Z$. Assume a linear regression model $f_\beta(\mathbf{x}) = \beta_1 \cdot \mathbf{x}_1 + \beta_2 \cdot \mathbf{x}_2 + \beta_3 \cdot \mathbf{x}_3$ trained from a sample data from $\mathbf{X} \times Y$ with $\beta_1, \beta_2 \approx 1$ and $\beta_3 \approx 0$. Intuitively, the inclusion of $\mathbf{X}_3$ in the model is due to an unclear understanding of which of the features contribute to the target feature. It turns out that $\mathbf{X} \not\perp Z$ but, for $\beta_3 = 0$ (which can be obtained by some fairness regularization method (Kamishima et al., 2011)), we have $f_\beta(\mathbf{X}) = \beta_1 \cdot \mathbf{X}_1 + \beta_2 \cdot \mathbf{X}_2 \perp Z$. It turns out (see Lemma B.1) that $\mathcal{S}(f_\beta, X) = (\beta_1 \cdot \mathbf{X}_1, \beta_2 \cdot \mathbf{X}_2, 0)$ and then $\mathcal{S}(f_\beta, \mathbf{X}) \perp Z$. This holds both in the interventional and in the observational variants.

The above represents an example where the input data depends on the protected feature, but the model and the explanations are independent from it.

### 4.2 Equal Treatment Inspection via Explanation Disparity

#### 4.2.1 Statistical Independence Test via Classifier AUC Test

In this subsection, we formalize a statistical test of independence based on the AUC of a binary classifier. The test of $\mathbf{W} \perp Z$ is stated in general form for multivariate random variables $\mathbf{W}$ and a binary random variable $Z$ with $dom(Z) = \{0, 1\}$. In the next subsection, we will instantiate it to the case $\mathbf{W} = \mathcal{S}(f_\theta, \mathbf{X})$.

Let $\mathcal{D} = \{(\mathbf{w}_i, z_i)\}_{i=1}^n$ be a dataset of realizations of the random sample $(\mathbf{W} \times Z)^n \sim \mathcal{F}^n$ where $\mathcal{F}$ is unknown. The independence $\mathbf{W} \perp Z$ can be tested via a two-sample test. In fact, we have $\mathbf{W} \perp Z$ iff $P(\mathbf{W}|Z) = P(\mathbf{W})$ iff $P(\mathbf{W}|Z = 1) = P(\mathbf{W}|Z = 0)$. We test whether the positive and negative instances in $\mathcal{D}$ are drawn from the same distribution by a classifier two-sample test, which does not rely on permutation of data nor it assumes equal proportion of positive and negatives as in (Lopez-Paz and Oquab, 2017, Sections 2 and 3). We rely on a probabilistic classifier $f : \mathbf{W} \to [0, 1]$, for which $f(\mathbf{w})$ estimates $P(Z = 1|\mathbf{W} = \mathbf{w})$, and on its AUC:

$$AUC(f) = \tag{2}$$
$$E_{(\mathbf{w}, Z), (\mathbf{W}', Z') \sim \mathcal{F}}[I((Z - Z') \cdot (f(\mathbf{W}) - f(\mathbf{W}')) > 0)]$$
$$+ \frac{1}{2} \cdot I(f(\mathbf{W}) = f(\mathbf{W}'))|Z \neq Z']$$

Under the null hypothesis $H_0 : \mathbf{W} \perp Z$, no classifier $f$ can be better than random guessing.

**Lemma 4.2.** *If $\mathbf{W} \perp Z$ then $AUC(f) = \frac{1}{2}$ for any $f$.*

*Proof.* Recall the definition of the Bayes Optimal classifier $f_{opt}(\mathbf{w}) = P(Z = 1|\mathbf{W} = \mathbf{w})$. For any classifier $f$:

$$AUC(f_{opt}) \geq AUC(f) \geq 1 - AUC(f_{opt}) \tag{3}$$

The first bound $AUC(f_{opt}) \geq AUC(f)$ follows because the Bayes Optimal classifier minimizes the Bayes risk (Gao and Zhou, 2015). Assume the second bound does not hold, i.e., for some $f$ we have $AUC(f_{opt}) < 1 - AUC(f)$. Consider the classifier $\bar{f}(\mathbf{w}) = 1 - f(\mathbf{w})$. We have $AUC(\bar{f}) \geq 1 - AUC(f)$, and then $\bar{f}$ would contradict the first bound because $AUC(f_{opt}) < AUC(\bar{f})$.

If $\mathbf{W} \perp Z$, then $P(Z = 1|\mathbf{W} = \mathbf{w}) = P(Z = 1)$, and then $f_{opt}(\mathbf{w})$ is constant. By (2), this implies $AUC(f_{opt}) = \frac{1}{2}$. By (3), this implies $AUC(f) = \frac{1}{2}$ for any classifier $f$. □

As a consequence, any statistics to test $AUC(f) = \frac{1}{2}$ can be used for testing $\mathbf{w} \perp Z$. A classical choice is to resort to the Wilcoxon–Mann–Whitney test, as done by Chakravarti et al. (2023), which, however, assumes that the distributions of scores for positives and negatives have the same shape. Alternatives with a better power include the Brunner–Munzel test (Neubert and Brunner, 2007) and the Fligner–Policello test (Fligner and Policello, 1981). The former is preferable, and it will be our choice in experiments, as the latter assumes that the distributions are symmetric.

### 4.2.2 Testing for Explanation Disparity via the Equal Treatment Inspector

We instantiate the previous AUC-based method for testing independence to the case of testing for Explanation Disparity via the Equal Treatment Inspector.

**Theorem 4.3.** *Let $g_\psi : \mathcal{S}(f_\theta, \mathbf{X}) \to [0, 1]$ be an "Equal Treatment Inspector" for the model $f_\theta$, and $\alpha$ a significance level. We can test the null hypothesis $H_0 : \mathcal{S}(f_\theta, \mathbf{X}) \perp Z$ at $100 \cdot (1 - \alpha)\%$ confidence level using a test statistics of $AUC(g_\psi) = \frac{1}{2}$.*

*Proof.* Under $H_0$, by Lemma 2 with $\mathbf{W} = \mathcal{S}(f_\theta, \mathbf{X})$ and $f = g_\psi$, we have $AUC(g_\psi) = \frac{1}{2}$. □

Results of the test can report the $p$-value of the adopted test for $AUC(g_\psi) = \frac{1}{2}$, or the confidence interval for $AUC(g_\psi)$, as returned by the Brunner–Munzel test or by the methods (DeLong et al., 1988; Cortes and Mohri, 2004; Gonçalves et al., 2014). $AUC(g_\psi)$, or its confidence intervals, can be used to quantify the Explanation Disparity unfairness.

### 4.2.3 Explanation Capabilities of Equal Treatment Inspector

The following example showcases one of our main contributions: detecting *the sources* of Explanation Disparity through interpretable by-design inspectors. Here, we assume that the model is linear. In the Appendix G.4, we will experiment with non-linear models.

**Example 4.3.** Let $\mathbf{X}$ be three independent features such that $E[\mathbf{X}_1] = E[\mathbf{X}_2] = E[\mathbf{X}_3] = 0$, and $\mathbf{X}_1, \mathbf{X}_2 \perp Z$, and $\mathbf{X}_3 \not\perp Z$. Given a random sample of i.i.d. observations from $\mathbf{X} \times Y$, a linear model $f_\beta(\mathbf{x}) = \beta_0 + \beta_1 \cdot \mathbf{x}_1 + \beta_2 \cdot \mathbf{x}_2 + \beta_3 \cdot \mathbf{x}_3$ can be built by OLS (Ordinary Least Square) estimation, possibly with $\beta_1, \beta_2, \beta_3 \neq 0$. It turns out (see Lemma B.1) that $\mathcal{S}(f_\beta, \mathbf{x})_i = \beta_i \cdot \mathbf{x}_i$. Consider now a linear ET Inspector $g_\psi(\mathbf{s}) = \psi_0 + \psi_1 \cdot \mathbf{x}_1 + \psi_2 \cdot \mathbf{s}_2 + \psi_3 \cdot \mathbf{s}_3$, which can be written in terms of the $\mathbf{x}$'s as: $g_\psi(\mathbf{x}) = \psi_0 + \psi_1 \cdot \beta_1 \cdot \mathbf{x}_1 + \psi_2 \cdot \beta_2 \cdot \mathbf{x}_2 + \psi_3 \cdot \beta_3 \cdot \mathbf{x}_3$. By OLS estimation properties, we have $\psi_1 \approx cov(\beta_1 \cdot \mathbf{X}_1, Z)/var(\beta_1 \cdot \mathbf{X}_1) = cov(\mathbf{X}_1, Z)/(\beta_1 \cdot var(\mathbf{X}_1)) = 0$ and analogously $\psi_2 \approx 0$. Finally, $\psi_3 \approx cov(\mathbf{X}_3, Z)/(\beta_3 \cdot var(\mathbf{X}_3)) \neq 0$. In summary, the coefficients of $g_\psi$ provide information about which feature contributes (and how much it contributes) to the dependence between the explanation $\mathcal{S}(f_\beta, \mathbf{X})$ and the protected feature $Z$. Notice that $f_\beta(\mathbf{X}) \not\perp Z$ as well, but we cannot explain which features contribute to such a dependence by looking at $f_\beta(\mathbf{X})$, since $\beta_i \approx cov(\mathbf{X}_i, Y)/var(\mathbf{X}_i)$ can be non-zero also for $i = 1, 2$.

## 5 Experimental Evaluation

We conduct experiments on equal treatment by systematically varying the model $f$, its parameters $\theta$, and the input data distributions $\mathcal{D}_\mathbf{X}$. We complement experiments described in this section by adding further experimental results in the Appendix that: *(i)* compare the different types of Shapley values estimation (Appendix E), *(ii)* add experiments on natural datasets (Appendix F), *(iii)* exhibit a larger range of modeling choices (Appendix G.3), *(iv)* compare AUC vs accuracy for the C2ST independence test (Appendix G.1), *(v)* extend the comparison against DP (Appendix G.5), and *(vi)* include LIME as an explanation method (Appendix H).

We adopt `xgboost` (Chen and Guestrin, 2016) for the model $f_\theta$, and a logistic regression for the inspector. We compare the AUC performances of several inspectors: $g_\psi$ (see Eq. 1) for ED (see Def. 3.2), $g_v$ for DP (see Def. 2.1), $g_\Upsilon$ for fairness of the input (i.e., $\mathbf{X} \perp Z$ as discussed in Section 4.1), and a combination $g_\phi$ of the last two inspectors to test $(f_\theta(\mathbf{X}), \mathbf{X}) \perp Z$. These are the formal definitions:

$$\Upsilon = \arg\min_{\tilde{\Upsilon}} \sum_{\mathbf{x} \in \mathcal{D}_{\mathbf{X} \setminus Z}^{val}} \ell(g_{\tilde{\Upsilon}}(\mathbf{x}), z_\mathbf{x}) \quad \upsilon = \arg\min_{\tilde{\upsilon}} \sum_{\mathbf{x} \in \mathcal{D}_{\mathbf{X} \setminus Z}^{val}} \ell(g_{\tilde{\upsilon}}(f_\theta(\mathbf{x})), z_\mathbf{x})$$

$$\phi = \arg\min_{\tilde{\phi}} \sum_{\mathbf{x} \in \mathcal{D}_{\mathbf{X} \setminus Z}^{val}} \ell(g_{\tilde{\phi}}(f_\theta(\mathbf{x}), \mathbf{x}), z_\mathbf{x})$$

### 5.1 Experiments with Synthetic Data

We generate synthetic datasets by first drawing $10,000$ samples from normally distributed features $\mathbf{X}_1 \sim N(0,1), \mathbf{X}_2 \sim N(0,1), (\mathbf{X}_3, \mathbf{X}_4) \sim N\left(\begin{bmatrix} 0 \\ 0 \end{bmatrix}, \begin{bmatrix} 1 & \gamma \\ \gamma & 1 \end{bmatrix}\right)$. Then, we define a binary protected feature $Z$ with values $Z = 1$ if $\mathbf{X}_4 > 0$ and $Z = 0$ otherwise. We compare the methods and baselines while varying the correlation $\gamma = r(\mathbf{X}_3, Z)$ from 0 to 1. We define two experimental scenarios below. In both of them, the model $f_\beta$ is a function over the domain of the features $\mathbf{X}_1, \mathbf{X}_2, \mathbf{X}_3$ only.

**Indirect Case:** *Unfairness in the data and in the model.* We consider all of the three features in the dataset $\mathbf{X}_1, \mathbf{X}_2, \mathbf{X}_3$. This gives rise to unfairness of the input parameterized by $\gamma = r(\mathbf{X}_3, Z)$. To generate DP violation in the model, we create the target $Y = \sigma(\mathbf{X}_1 + \mathbf{X}_2 + \mathbf{X}_3)$, where $\sigma$ is the logistic function.

**Uninformative Case:** *Unfairness in the data and fairness in the model.* The unfairness in the input data remains the same as in the previous case, while we now remove unfairness in the model. The target feature is now defined as $Y = \sigma(\mathbf{X}_1 + \mathbf{X}_2)$. The $\gamma$ parameter controls unfairness in the dataset, which should not be captured by the model, since $\mathbf{X}_1, \mathbf{X}_2 \perp Z$ implies $Y \perp Z$ by propagation of independence.

In Figure 2, we compare the AUC performances of the different inspectors on synthetic data split into $1/3$ for training the model, $1/3$ for training the inspectors and $1/3$ for testing them. Overall, the ED Inspector $g_\psi$ is able to detect unfairness in both scenarios. The DP inspector $g_v$ works fine in the uninformative case, but for the indirect, it is less sensitive than the other methods. Finally, the inspectors $g_\Upsilon$ and $g_\phi$ detect unfairness in the input but not in the model. Further experiments are shown in Appendix G.4 to investigate the contribution of the explanation distribution features, namely the $\mathcal{S}(f_\theta, \mathbf{x})_i$'s, to the ED Inspector $g_\psi$.

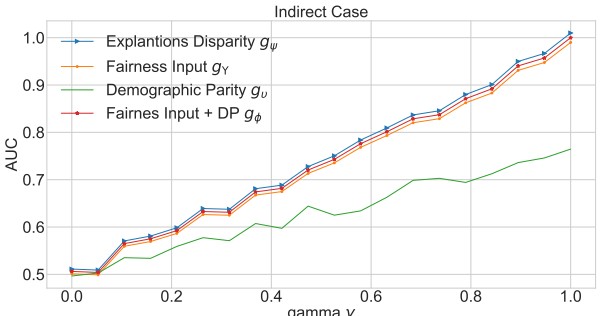 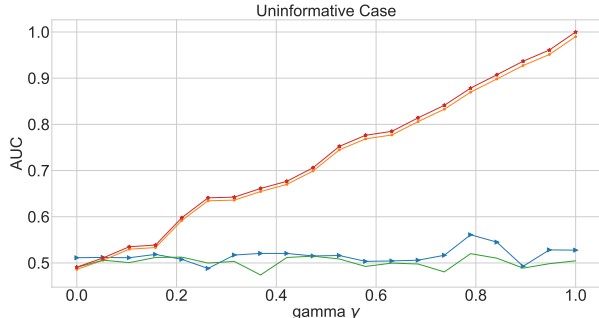

Figure 2: In the "Indirect case" (left): good unfairness detection methods should follow an increasing steady slope to capture the fairness violation; the DT inspector appears less sensitive due to the low dimensionality of its input. In the "Uninformative case" (right): good unfairness detection methods should remain constant with an AUC $\approx 0.5$; the inspectors based on input data ($g_\Upsilon$ and $g_\phi$) flag a false positive case of unfairness.

## 5.2 ACS Travel Time Data

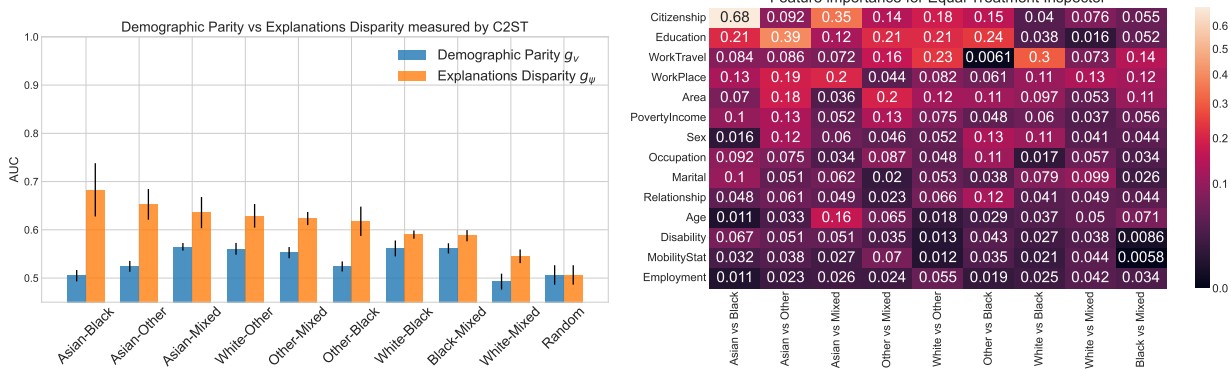

Figure 3: The left figure compares ED and DP measures on the US Travel Time data. The AUC range for ED is notably wider, and aligning with the theoretical section, there are instances where DP fails to identify discrimination that ED successfully detects. "Random" refers to the scenario where two disjoint groups were created by random sampling from the overall population. For a detailed statistical analysis, please refer to Appendix G.5. The right figure provides insight into the influential features contributing to unequal treatment. Higher feature values correspond to a greater likelihood of these features being the underlying causes of unequal treatment.

We experiment here with the ACS Travel Time dataset (Ding et al., 2021), and in the Appendix F with three other ACS datasets. The fairness notions are tested against all pairs of groups from the protected attribute "Race". Figure 3 (left) shows the AUC performances of the ED Inspector $g_\psi$ and the DP inspector $g_v$. The standard deviation of the AUC is calculated over 30 bootstrap runs, each one splitting the data into 1/3 for training the model, 1/3 for training the inspectors and 1/3 for testing them. The AUCs for the ED Inspectors are greater than the ones for the DP inspectors, which is expected due to Lemmma 4.1. In the Appendix G.1, the results of the C2ST tests are discussed.

Figure 3 (right) shows the Wasserstein distance between the coefficients of the linear regressor $g_\psi$ compared to a baseline where groups are assigned at random in the input dataset. The mean orders the matrix in both dimensions. This feature importance post-hoc explanation method provides insights into the impact of different features as sources of unequal treatment. We observe "Education" as a highly discriminatory proxy of ethnicity while the role of the feature "Employment" is less relevant. This allows us to identify areas where adjustments or interventions may be needed to move closer to the ideal of equal treatment.

## 6    Conclusions

The policy of *equal treatment as blindness*, often referred to in machine learning as fairness as unawareness, has long been critiqued by social scientists. In this work, we argue that existing fairness metrics fail to adequately address these criticisms. To address this gap, we introduce a novel metric that aims to more effectively capture these concerns. While related work applied Demographic Parity, we have provided theoretical and experimental evidence that this fairness metric does not adequately measure fairness. Our notion of Explanation Disparity is more fine-grained, accounting for the usage of attributes by the model via explanation distributions. Consequently, Explanation Disparity implies Demographic Parity, but the converse is not necessarily true, which we confirmed theoretically and experimentally.

This paper seeks to improve the understanding of how theoretical concepts of fairness from liberalism-oriented political philosophy align with technical measurements. Rather than merely comparing one social group to another based on disparities within decision distributions, our metric of Explanation Disparity considers differences through the explanation distribution of all non-protected attributes, which often act as proxies for protected characteristics. Implications warrant further techno-philosophical discussions.

**Limitations:** Political philosophical notions and policies are more complex than we can account for in this paper. Our research has focused on tabular data using Shapley values, which allow for theoretical guarantees but may differ from their computational approximations. It is possible that alternative AI explanation techniques, such as feature attribution methods, logical reasoning, argumentation, or counterfactual explanations, could be useful and offer their unique advantages to definitions of equal treatment. It is important to note that employing fair-ML techniques does not necessarily ensure fairness in socio-technical systems based on AI, as stated in Kulynych et al. (2020). See impact statement in Appendix I for further information and implications.

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

## Contents

## A  Notions of Fairness: from Philosophical Foundations to Implications in Practice

In this section, we present the philosophical intuitions behind different notions of fairness illustrated with an applied case study.[1] Next, we collect the technical requirements that fairness metrics must meet to align with *equal treatment as blindness* criticism and to be viable for the practical applicability of fairness metrics in real-world scenarios where data constraints and accountabilty on the results are often important. Finally, we conduct a broader survey of fairness metrics and relate them to these requirements, examining how each metric addresses the alignment and practical challenges.

### A.1  An Overview of Notions of Fairness in Distributive Justice

To illustrate the difference between equal opportunity, equal outcomes, and equal treatment, we consider the hypothetical use case of the admission process of a university.

In June 2023, the U.S. Supreme Court struck down race-conscious admission programs at some universities (Killenbeck and Killenbeck, 2022)[2], ruling these as discriminatory against certain racial groups. This decision marked a significant shift from previous policies that used race as one of several factors in a holistic admissions process to promote a diverse student body[3]. While the ruling does not completely forbid universities from considering applicants' racial features, it emphasizes a more colorblind approach to admissions. This move towards equal treatment focuses less on achieving equal outcomes (diverse representation) and more on not considering race as an admissions factor. Now we discuss these notions within the context of fair machine

---

[1]We do not claim to cover comprehensively different philosophical approaches to such a case study. While a comprehensive philosophical treatise may require a whole book, we only want to illustrate some issues, on which we are basing our arguments.

[2]https://www.scotusblog.com/2023/06/supreme-court-strikes-down-affirmative-action-programs-in-college-admissions/

[3]https://www.politico.com/news/2023/06/29/supreme-court-ends-affirmative-action-in-college-admissions-00104179

learning. We now compare to the different notions from distributive justice commonly discussed in the literature.

*Equal Outcomes.* This notion requires that the distribution of acceptance rates is similar, independently of the student (cf. Definition 2.1). This could mean setting targets or quotas to ensure representation from various groups, regardless of their individual academic credentials. For instance, if 20% of applicants are from a certain ethnicity, the university might aim to have 20% of their admits from that ethnicity as well. While this promotes a diverse student body, it can overlook individual merit and potentially lead to tension between equity and academic standards. Particularly given the recent ruling of the US Supreme Court, considering ethnicity to match the proportion of applicants from that ethnicity would likely be considered illegal[4]. It's also important to note that outcomes can have similar rates due to random chance.

*Equal Opportunity.* In this approach, the university would ensure that students from different backgrounds have an equal chance of being admitted if they have high potential or qualifications. For instance, students from under-resourced high schools are given the same opportunity as those from well-funded schools if they demonstrate the same exceptional talent or potential. The university may implement this strategy by adjusting admission criteria based on educational opportunities tied to race. Policies following this notions can lead to challenges like overcompensating for disadvantages or unintentionally lowering standards for certain groups.

*Equal Treatment.* In this scenario, the university ensures that every application is evaluated based on uniform criteria such as academic achievement, extracurricular involvement, and personal essays, without bias towards the applicant's background. This approach means that factors like a student's socioeconomic status, high school's reputation, geographic location, or other attributes not chosen at birth (protected attributes) would not directly influence the admission procedure.

In the university admission case, each approach to fairness in admissions — equal outcome, equal opportunity, and equal treatment — has its merits and issues. Equal outcomes, strive for a representative student body, assuming that all groups have inherently the same academic capabilities. Equal opportunity aims to level the playing field based on the potential of the candidates. Equal treatment focuses on ignoring protected attributes aiming to achieve a uniform and unbiased evaluation process. We leave the normative discussion of which fairness paradigm should be pursued by policy to the discourse in the social sciences and the broad public.

## A.2 Requirements Collection

If we aim to measure the impact of *equal treatment as blindness* policy in the scenario involving college admissions, liberal-oriented political scientists argue that the university should ensure that every application is evaluated based on the same criteria, such as academic achievement, extracurricular involvement, and personal essays, without bias towards the applicant's background. This means that a student's protected attributes, such as ethnicity, socioeconomic status, or geographic location, should not influence their likelihood of admission. From an application standpoint, this approach also necessitates practical measures to ensure that the systems can be effectively monitored and maintained once deployed.

The challenge here is to ensure the following in the admissions process:

**(R1) Group Discrimination:** refers to making decisions based on inherent or protected characteristics of individuals, such as race, gender, or ethnicity. This in in line with all three notions of equality, and policies should ensure that individuals are not discriminated against based on characteristics beyond their control (protected attributes)(Hertweck et al., 2024).

**(R2) Unlabeled Data:** Two common ways to evaluate notions of equality are on the decisions, or on the consequences of the decisions (Driver, 2022). This criterion evaluates discriminatory behaviour on the model's decision instead of the statistical differences in errors. It was derived to align with the intuition of disparate treatment of the model and measured if the metric looks at $f_\theta(X)$ instead of disparate impact $f_\theta(X) - Y$

---

[4] https://www.jdsupra.com/legalnews/the-impact-of-the-supreme-court-s-7330075/

***(R3) No Background Knowledge***: Fairness metrics should not require an understanding of causal or structural aspects of the data. This ensures that the metrics are practical and can be easily applied in real-world scenarios where detailed background knowledge may be lacking. This criterion is essential for making fairness assessments accessible to practitioners who may not have specialized knowledge about the underlying data structures or causal relationships.

***(R4) Proxy discrimination***: Even if the protected attribute is not used directly in the decision-making process, there might be features that have a dependency on it, this phenomenon is known as proxy discrimination (Wachter et al., 2020).This criterion ensures that metrics can detect if the model's behavior is genuinely free from discriminatory features arising from proxy discrimination. It was derived from statements such as the one in the introductionMinow (1990), to scrutinize the impact of various features on predictions and prevent indirect biases. This requirement is measured if the metric has theoretical properties and guarantees, such as independence, as we derive in Section 4.2.1.

***(R5) Explanation Capabilities***: This focuses on the need for fairness metrics to not only detect biases or discrimination but also to explain them. This criterion is derived from technical requirement to improve transparency and understanding of how and why biases occur in model predictions. This requirement is met if a metric has theoretical evidence, as we derive in Section 4.2.3, or provides software that allows for empirical evaluation, as we provide in the Python package `explanationspace`.

### A.3   Survey of Fairness Metrics and their Relationship to Requirements

In this section, we compare our requirements to measure the impact of the policy of *equal treatment as blindness* with respect to other metrics found in the literature. We distinguish between two discrimination types: *disparate treatment*, aiming to prevent intentional model discrimination, and *disparate impact*, addressing unintentional effects across subpopulations(Commission, 1964; Kuppler et al., 2021). The former examines the system's intentions, as realized in model predictions $f_\theta(\mathbf{X})$, while the latter considers effects, interpreted as the error $f_\theta(\mathbf{X}) - Y$.

### A.3.1   Disparate Impact Fairness Metrics: Approaches that Rely on Labeled Data

A notable technical limitation of disparate impact fairness metrics is that they necessitate labelled data for computation. Acquiring labelled data post-deployment poses a considerable challenge and is often impractical, exhibiting well-known biases, such as confounding, selection, and missingness (Feng et al., 2023; Ruggieri et al., 2023). Thus, none of the following metrics meet the criterium *(R2) Unlabelled data* because they require labeled data.

**Equal Opportunity**   Two formulations of the Equal Opportunity fairness metric are defined in the literature. First, it is the difference in the True Positive Rate (TPR) between the protected group and the reference group (Hardt et al., 2016; Podesta et al., 2014; Munoz et al., 2016):

$$\text{TPR} = \frac{TP}{TP + FN}$$
$$\text{EOF}_z = \text{TPR}_z - \text{TPR}$$

Second – a formulation proposed by Heidari et al. (2019) and that does not rely on the false negatives – a model $f_\theta$ achieves equal opportunity if $P(f_\theta(\mathbf{X}) = 1 | Y = 1, Z = z) = P(f_\theta(\mathbf{X}) = 1 | Y = 1)$.

Equal Opportunity comes with a technical limitation, as it necessitates the availability of labeled data for true positives, hence it does not meet *(R2)*. From the philosophical alignment perspective, both metrics may align better with egalitarian ideals, as they prioritize access to good outcomes through error ratios.

In contrast, our conceptualization of equal treatment diverges in its alignment, finding resonance with distributive justice principles rooted in liberal ideology and, in this particular requirement, not necessitating labelled data.

**Treatment Equality** Despite its similar name, Treatment Equality does not necessarily align with the same distributive justice values as equal treatment, and the mathematical metric it employs is notably distinct (Berk et al., 2021):

$$\frac{FP_z}{FN_z} = \frac{FP}{FN}$$

From a technical perspective, the challenge relies on obtaining false positive and false negative data, therefore not meeting the requirement *(R2)* on labelled data.

The authors do not specify the principle or value with which the notion should align, and this clarity is absent from their definition. An illustrative sentence is "*men and women are being treated differently by the algorithm*". However, this statement may not be entirely accurate, as the algorithm may treat men and women equally while exhibiting different error rates. For instance, a constant classifier that grants loans to everyone treats everyone equally. Still, if males and females have distinct patterns of loan repayment, the classifier will yield different errors, resulting in distinct $\frac{FP}{FN}$ values. Then this metric of "treament equality" is very distinct from our proposed notion due to not belonging to a clear distributive justice perspective and the technical implementation.

### A.3.2 Disparate Treatment Fairness Metrics: Approaches that Determine Unfairness without Labelled Data

In this section we compare against metrics that rely only on $\mathbf{X}$ and $f_\theta$ rather than on disparities of model errors. Therefore they all meet the *(R2) Unlabelled data* requirement.

**Fairness Through Unawareness.** This bias mitigation method was initially proposed by Grgic-Hlaca et al. (2018) and is often used as a baseline.

**Definition A.1.** *(Fairness Through Unawareness).* An algorithm is fair if no protected attributes $Z$ is explicitly used as an input of the algorithm.

Any mapping $f : \mathbf{X} \to Y$ for which $Z \notin \mathbf{X}$ satisfies such a definition. It has a clear shortcoming as features in $\mathbf{X}$ can contain discriminatory information, known as proxy features for discrimination. Further, the research requirement of detecting *(R4) Proxy discrimination* is not met. One of the main contributions requirements for the notion of equal treatment and technical contributions of Explanation Disparitiy is that it captures statistical relations of all the features with the protected attribute. In our approach, the model does not consider the protected attribute to be present in the covariates $X$, therefore implying fairness through unawareness. Furthermore, in Section 4.1 of the main body, we discuss the limitations of analyzing the Shapley values of the protected attribute.

**Demographic Parity.** To define a quality metric, one must collect domain requirements and formalise the metric such that its mathematical properties match the requirements. The domain requirements in this paper are given by philosophical foundations (cf. Appendix A). From the mathematical perspective, we have shown that Explanation Disparity is a more aligned notion to *equal treatment* than Demographic Parity to the liberal notion of *equal treatment.*

A refined metric introduced by legal scholars aiming to align with the European Court of Justice "gold standard" (Wachter et al., 2021) is Conditional Demographic Parity, which extends Demographic Parity by fixing one or more attributes.

**Definition A.2.** *(Conditional Demographic Parity (CDP)).* A model $f_\theta$ achieves conditional demographic parity if $P(f_\theta(\mathbf{X})|\mathbf{X}_i = \tau, Z = z) = P(f_\theta(\mathbf{X})|\mathbf{X}_i = \tau)$, where $\tau$ is any fixed value.

CDP differs from DP only in the sense that one or more additional covariate conditions are added. CDP, therefore, is not able to detect proxy discrimination *(R4)* nor capable to account for the sources behind discrimination *(R5)*. Our notion for Explanation Disparity aims to align further with liberal-oriented equal treatment, and this is only achieved if the contributions of all features to the prediction are equal for each of the protected subgroups.

**Individual Fairness.** We say that a model $f$ achieves individual fairness if similar individuals receive similar predictions:

$$\forall \mathbf{x}, \mathbf{x}' \; d(f(\mathbf{x}), f(\mathbf{x}')) \leq \mathcal{L} \cdot d(\mathbf{x}, \mathbf{x}')$$

Where $\mathcal{L} > 0$ is a constant and the two $d(\cdot, \cdot)$'s are distance functiosn between models' outputs and between instances respectively (Dwork et al., 2012).

From a philosophical perspective, individual fairness is closest to the liberal point of view. However, it fails the requirements of liberalist arguments. Liberalism argues for meritocracy, i.e., disparate treatment may be considered fair if it is based on varying efforts or preferences of individuals but unfair if it is based on protected characteristics that individuals did not choose. E.g., it's fair to hire someone because of better grades, but it is unfair if these grades depend on ethnicity. The definition of individual fairness does not consider protected characteristics or proxies thereof failing research requirement *(R1)* of group discrimination.. While individual fairness may be easily combined with blindness or adjusted definitions of similarities (Fleisher, 2021; Yurochkin et al., 2020), the treatment of proxies to protected characteristics is hard to avoid, rendering the alignment of an AI model with distributive justice values difficult. Therefore, individual fairness is not a popular metric — unlike group fairness metrics (Fleisher, 2021).

**Counterfactual Fairness.** Counterfactual fairness, as defined by Kusner et al. (2017) captures the intuition that a decision is fair towards an individual if it is the same in *(i)* the actual case and *(ii)* in a counterfactual case where the individual belonged to a different protected attribute group.

**Definition A.3.** *Counterfactual Fairness* We say that the model $f_\theta$ is counterfactually fair if:

$$\forall \mathbf{x} \in \mathbf{X} \; \forall z, z' \in Z \; P(f_\theta(\mathbf{x}_{Z=z}) | \mathbf{X} = \mathbf{x}, Z = z) = P(f_\theta(\mathbf{x}_{Z=z'}) | \mathbf{X} = \mathbf{x}, Z = z)$$

where $\mathbf{x}_{Z=z}$ and $\mathbf{x}_{Z=z'}$ are the instances $\mathbf{x}$ in the (counterfactual) worlds where $Z = z$ and $Z = z'$ respectively.

Rosenblatt and Witter (2023) have shown that: (i) an algorithm that satisfies counterfactual fairness also satisfies demographic parity; and (ii) all algorithms that satisfy demographic parity can be modfied to satisfy counterfactual fairness. These results conclude that counterfactual fairness is equivalent to demographic parity, therefore failing the same requirements *(R4)*, proxy discrimination and *(R5)* explanation capabilities. Moreover, this fairness definition fails on our requirement *(R3)*, of not needing background knowledge – since counterfactuals can only be computed for a given causal graph.

**Decomposition Method Specific of Demographic Parity.** We formally compare our approach to the prior work of Lundberg (2020) and the related SHAP Python package documentation. The authors addressed DP using SHAP value estimation. This brief workshop paper emphasizes the importance of "decomposing a fairness metric among each of a model's inputs to reveal which input features may be driving any observed fairness disparities". In terms of statistical independence, the approach can be rephrased as decomposing $f_\theta(\mathbf{X}) \perp Z$ by examining $\mathcal{S}(f_\theta, \mathbf{X})_i \perp Z$ individually for $i \in [1, p]$. Actually, the paper limits itself to consider only differences of means, namely testing for $E[\mathcal{S}(f_\theta, \mathbf{X})_i | Z = 1] \neq E[\mathcal{S}(f_\theta, \mathbf{X})_i | Z = 0]$. However, the method is not sufficient nor necessary to prove DP, as shown next.

**Lemma A.4.** $f_\theta(\mathbf{X}) \perp Z$ *is neither implied by nor it implies* $(\mathcal{S}(f_\theta, \mathbf{X})_i \perp Z \text{ for } i \in [1, p])$.

*Proof.* Consider $f_\theta(\mathbf{X}) = \mathbf{X}_1 - \mathbf{X}_2$ with $\mathbf{X}_1, \mathbf{X}_2 \sim \text{Ber}(0.5)$ and $Z = 1$ if $\mathbf{X}_1 = \mathbf{X}_2$, and $Z = 0$ otherwise. Hence $Z \sim \text{Ber}(0.5)$. We have $\mathcal{S}(f_\theta, \mathbf{X})_1 = \mathbf{X}_1 \perp Z$ and $\mathcal{S}(f, \mathbf{X})_2 = -\mathbf{X}_2 \perp Z$. However, $f_\theta(\mathbf{X}) \not\perp Z$, e.g., $P(Z = 0 | f_\theta(\mathbf{X}) = 0) = 1 \neq 0.5 = P(Z = 0)$. Example 4.1 illustrates a case where $f_\theta(\mathbf{X}) \perp Z$ yet $\mathcal{S}(f_\theta, \mathbf{X})_1$ and $\mathcal{S}(f_\theta, \mathbf{X})_2$ are not independent of $Z$. $\square$

Our metric of Explanation Disparity considers the independence of the *multivariate* distribution of $\mathcal{S}(f, \mathbf{X})$ with respect to $Z$, rather than the independence of each marginal distribution $\mathcal{S}(f_\theta, \mathbf{X})_i \perp Z$. With our definition, we obtain a sufficient condition for DP, as shown in Lemma 4.1.

Chang et al. (2023) follows a similar approach but from the perspective of discovering which subgroup $z \in Z$ exhibits the largest importance disparity relative to a feature $\mathbf{X}_i$.

**Definition A.5.** *(Feature Importance Disparity)* Assume $Z$ is discrete, not necessarily binary. The feature importance disparity of $z \in Z$ relatively to a feature $\mathbf{X}_i$ is defined as:

$$\mathtt{FID}(z, i) = |E[\mathcal{S}(f_\theta, \mathbf{X})_i | Z = z] - E[\mathcal{S}(f_\theta, \mathbf{X})_i]|$$

From an alignment perspective, the authors of this method don't clarify which philosophical equality criteria the proposed metric measures. Moreover, assuming DP as the reference notion, the method shares the same pitfalls illustrated by the lemma above for Lundberg (2020). From the requirements perspective, the definition does not meet *(R4)* proxy discrimination detection due to using the expected value to measure distribution differences.

## B  Definition and Properties of Shapley Values

Explainability has become an important concept in legal and ethical requirements of machine learning applications (Selbst and Barocas, 2018). A wide variety of methods have been developed, aiming to account for the decision of algorithmic systems (Guidotti et al., 2019; Mittelstadt et al., 2019; Arrieta et al., 2020). One of the most popular approaches to explainability in machine learning is based on Shapley values.

Shapley values are used to attribute relevance to features according to how the model relies on them (Lundberg et al., 2020; Lundberg and Lee, 2017; Rozemberczki et al., 2022). Shapley values are a coalition game theory concept that aims to allocate the surplus generated by the grand coalition in a game to each of its players (Shapley, 1997).

For set of players $N = \{1, \ldots, p\}$, and a value function $\mathrm{val} : 2^N \to \mathbb{R}$, the Shapley value $\mathcal{S}_i$ of the $i$'th player is defined as the average marginal contribution of player $i$ in all possibles coalitions of players:

$$\mathcal{S}_i = \sum_{T \subseteq N \setminus \{i\}} \frac{|T|!(p - |T| - 1)!}{p!}(\mathrm{val}(T \cup \{i\}) - \mathrm{val}(T))$$

In the context of machine learning models, players correspond to features $\mathbf{X} = (\mathbf{X}_1, \ldots, \mathbf{X}_p)$, and the contribution of the feature $\mathbf{X}_i$ is with reference to the prediction of a model $f$ for an instance $\mathbf{x}^\star$ to be explained. Thus, we write $\mathcal{S}(f, \mathbf{x}^\star)_i$ for the Shapley value of feature $\mathbf{X}_i$ in the prediction $f(\mathbf{x}^\star)$. We denote by $\mathcal{S}(f, \mathbf{x}^\star)$ the vector of Shapley values $(\mathcal{S}(f, \mathbf{x}^\star)_1, \ldots, \mathcal{S}(f, \mathbf{x}^\star)_p)$.

There are two variants for the term $\mathrm{val}(T)$ (Aas et al., 2021; Chen et al., 2020; Zern et al., 2023): the *observational* and the *interventional* one. When using the observational conditional expectation, we consider the expected value of $f$ over the joint distribution of all features conditioned to fix features in $T$ to the values they have in $\mathbf{x}^\star$:

$$\mathrm{val}(T) = E[f(\mathbf{x}^\star_T, \mathbf{X}_{N \setminus T}) | \mathbf{X}_T = \mathbf{x}^\star_T] \tag{4}$$

where $f(\mathbf{x}^\star_T, \mathbf{X}_{N \setminus T})$ denotes that features in $T$ are fixed to their values in $\mathbf{x}^\star$, and features not in $T$ are random variables over the joint distribution of features. Opposed, the interventional conditional expectation considers the expected value of $f$ over the marginal distribution of features not in $T$:

$$\mathrm{val}(T) = E[f(\mathbf{x}^\star_T, \mathbf{X}_{N \setminus T})] \tag{5}$$

In the interventional variant, the marginal distribution is unaffected by the knowledge that $\mathbf{X}_T = \mathbf{x}^\star_T$. In general, the estimation of (4) is difficult, and some implementations (e.g., SHAP) actually consider (5) as the default one. In the case of decision tree models, TreeSHAP offers both possibilities.

The Shapley value framework is the only feature attribution method that satisfies the properties of efficiency, symmetry, uninformativeness and additivity (Molnar, 2019; Shapley, 1997; Winter, 2002; Aumann and Dreze, 1974). We recall next the key properties of efficiency and uninformativeness.

**Efficiency.** Feature contributions add up to the difference of prediction for $\mathbf{x}^\star$ and the expected value of $f$:

$$\sum_{i \in N} \mathcal{S}(f, \mathbf{x}^\star)_i = f(\mathbf{x}^\star) - E[f(\mathbf{X})]) \tag{6}$$

The following property only holds for the interventional variant (e.g., for SHAP values), but not for the observational variant.

**Uninformativeness.** A feature $\mathbf{X}_i$ that does not change the predicted value (i.e., such that $f(\mathbf{x}) = f(\mathbf{x}')$ if $\mathbf{x}$ and $\mathbf{x}'$ differ only in the $i$-th element) has a Shapley value of zero, i.e., $\mathcal{S}(f, \mathbf{x}^\star)_i = 0$.

Finally, a full characterization of Shapley values can be given for linear models (see Aas et al. (2021)).

**Lemma B.1.** *Consider a linear model $f_\beta(\mathbf{x}) = \beta_0 + \sum_i \beta_i \cdot \mathbf{x}_i$. The Shapley values of the interventional variant turn out to be $\mathcal{S}(f, \mathbf{x}^\star)_i = \beta_i(\mathbf{x}_i^\star - \mu_i)$ where $\mu_i = E[\mathbf{X}_i]$. For the observational variant, this holds if the features are independent.*

## C  Detailed Related Work

This section provides an in-depth review of the related theoretical work that informs our research. We contextualize our contribution within the broader field of explainable AI and fairness auditing. We discuss the use of fairness measures such as demographic parity, as well as explainability techniques like Shapley values and counterfactual explanations.

### C.1  Explainability and Fair Supervised Learning

The intersection of fairness and explainable AI has been an active research topic in recent years. The work most close to our approach is Lundberg (2020) where Shapley values are aimed at testing for demographic parity (we provide a formal comparison in section A.3.2). Similarly, Chang et al. (2023) proposes a method that uses the expected feature importance and computes subgroups which maximize feature importance disparity. Their work differs with respect to Lundberg (2020) and our work, in that they look for finding those groups while our work assumes the groups as given.

Stevens et al. (2020) present an approach based on adapting the Shapley value function to explain model unfairness. They also introduce a new meta-algorithm that considers the problem of learning an additive perturbation to an existing model in order to impose fairness. In our work, we use the theoretical Shapley properties to provide fairness auditing guarantees. Our Equal Treatment Inspector is not perturbation-based but uses Shapley values to project the model to the explanation distribution, and then measures disparities. It also allows us to pinpoint the specific features driving such a violation.

Grabowicz et al. (2022) present a post-processing method based on Shapley values aiming to detect and nullify the influence of a protected attribute on the output of the system. For this, they assume there are direct causal links from the data to the protected attribute and that there are no measured confounders. Our work does not use causal graphs but exploits the theoretical properties of the Shapley values to obtain fairness model auditing guarantees.

Begley et al. (2020) propose a meta-algorithm for applying fairness interventions to already-trained models, wherein one trains a perturbation to the original model, rather than a new model entirely by adapting the Shapley value function w.r.t. demographic parity. In our work, we do not adopt the Shapley value function, but train a meta-algorithm in the testing phase, there is no need for labelled data, nor is our method based on perturbations.

Other lines of work have studied the relationship between explanation quality and population subgroups. For example,Dai et al. (2022), explores group-based disparities in explanation quality, focusing on metrics such as fidelity, stability, consistency, and sparsity. Aïvodji et al. (2019), introduces the concept of "fairwashing", where rule-based explanation methods can be manipulated to justify unfair model decisions, making them appear fair. Slack et al. (2020) demonstrate how machine learning algorithms can be deliberately designed so that explanation methods like LIME and SHAP obscure the model's explicitly unfair behavior, making it harder to detect biases. While these works focus on individual or local explanations, our research examines distributions of explanations, where individual differences might be masked by overall metric aggregation. For example, Figure 10 shows that SHAP and LIME yield similar results despite their distinct explanation mechanisms, and Appendix E analyzes various Shapley value approximations, which still produce comparable results. Investigating errors in calculating explanation distributions remains an avenue for future research.

Few works have researched fairness using other explainability techniques such as counterfactual explanations (Kusner et al., 2017; Manerba and Guidotti, 2021; Mutlu et al., 2022). We do not focus on counterfactual explanations but on feature attribution methods that allow us to measure unequal feature contribution to the prediction. Further work can be envisioned by applying explainable AI techniques to the Equal Treatment Inspector, or by constructing the explanation distribution out of other techniques.

## C.2 Classifier Two-Sample Test (C2ST)

The use of classifiers as a statistical tests of independence $\mathbf{W} \perp Z$ for a binary $Z$'s has been previously explored in the literature (Lopez-Paz and Oquab, 2017). The approach relies on testing accuracy of a classifier trained to distinguish $Z = 1$ (positives) from $Z = 0$ (negatives) given $\mathbf{W} = \mathbf{w}$. In the null hypothesis, i.e., the distributions of positives and negatives are the same, no classifier is better than a random answer with accuracy $1/2$. This assumes equal proportion of positive and negative instances. Our approach builds on this idea, but it considers testing the AUC instead of the accuracy. Thus, we remove the assumption of equal proportions[5]. We also show in Section G.1 that using AUC may achieve a better power than using accuracy. Chakravarti et al. (2023) have also used C2ST together with the AUC, with their implementation using bootstrap and asymptotic methods. Their work does not provide a formal proof as we do. Also, we use the Brunner–Munzel test, which has a better power; see Section G.1 for an experimental comparison.

Liu et al. (2020) propose a kernel-based approach to two-sample test classification. Alike work has also been used in Kaggle competitions under the name of "Adversarial Validation" (Ellis, 2023; Guschin et al., 2018; Barrabés et al., 2023), a technique which aims to detect which features are distinct between train and leaderboard datasets to avoid possible leaderboard shakes. Other works have integrated predicting the protected attribute in their pipeline. Edwards and Storkey (2016) remove statistical imparity from images by using an adversary that tries to predict the relevant sensitive variable from the model representation and censoring the learning of the representation of the model and data on images and neural networks. Our approach does not aim to censor or mitigate bias on the learned representation but to propose a measure of fairness that is more aligned with the liberal-oriented politics notion of equal treatment.

## D Explanation Disparity Given Shapley Values of Protected Feature

Can we measure *equal treatment* by looking only at the Shapley value of the protected feature? The following result considers a linear model (with unknown coefficients) over *independent* features. In such a very simple case, resorting to Shapley values leads to an exact test of both DP and ED, which turn out to coincide. Throughout this section, we assume an exact calculation of the Shapley values $\mathcal{S}(f_\theta, \mathbf{x})$ for instance $\mathbf{x}$, possibly for the observational and interventional variants – see (4,5) in Appendix B. In the following, we write $distinct(\mathcal{D}_\mathbf{X}, i)$ for the number of distinct values in the $i$-th feature of dataset $\mathcal{D}_\mathbf{X}$, and $\mathcal{S}(f_\beta, \mathcal{D}_\mathbf{X})_i \equiv 0$ if the Shapley values of the $i$-th feature are all 0's, i.e., if $\forall \mathbf{x} \in \mathcal{D}_\mathbf{X}.\mathcal{S}(f_\beta, \mathbf{x})_i = 0$.

**Lemma D.1.** *Consider a linear model* $f_\beta(\mathbf{x}) = \beta_0 + \sum_j \beta_j \cdot \mathbf{x}_j$. *Let $Z$ be the $i$-th feature, i.e., $Z = \mathbf{X}_i$, and let $\mathcal{D}_\mathbf{X}$ be such that $distinct(\mathcal{D}_\mathbf{X}, i) > 1$. If the features in $\mathbf{X}$ are independent, then $\mathcal{S}(f_\beta, \mathcal{D}_\mathbf{X})_i \equiv 0 \Leftrightarrow \mathcal{S}(f_\beta, \mathbf{X}) \perp Z \Leftrightarrow f_\beta(\mathbf{X}) \perp Z$.*

*Proof.* By Lemma B.1, it turns out $\mathcal{S}(f_\beta, \mathbf{x})_i = \beta_i(\mathbf{x}_i - E[\mathbf{X}_i])$. This holds both for the interventional variant, and, given the independence assumption, also for the observational variant. Since $distinct(\mathcal{D}_\mathbf{X}, i) > 1$, it turns out that $\mathcal{S}(f_\beta, \mathcal{D}_\mathbf{X})_i \equiv 0$ iff $\beta_i = 0$. Let us show $\beta_i = 0 \Rightarrow \mathcal{S}(f_\beta, \mathbf{X}) \perp Z \Rightarrow f_\beta(\mathbf{X}) \perp Z \Rightarrow \beta_i = 0$.

If $\beta_i = 0$, then $\mathcal{S}(f_\beta, \mathbf{X})$ consists of $\beta_j(\mathbf{x}_j - E[\mathbf{X}_j])$ in positions $j \neq i$, and of 0 in position $i$. Since the features in $\mathbf{X}$ are independent, by propagation of independence, $\mathcal{S}(f_\beta, \mathbf{X}) \perp Z$.

The implication $\mathcal{S}(f_\beta, \mathbf{X}) \perp Z \Rightarrow f_\beta(\mathbf{X}) \perp Z$ holds in general, as shown in Lemma 4.1.

Finally, by linearity of the model, $f_\beta(\mathbf{X}) \perp Z$ trivially implies $\beta_i = 0$. $\qquad\square$

---

[5]For unequal proportions, one can consider the accuracy of the majority class, but this still make the requirement to know the true proportion of positives and negatives.

However, the result does not extend to the case of dependent features.

**Example D.1.** Consider $Z = \mathbf{X}_2 = \mathbf{X}_1^2$, and the linear model $f_\beta(\mathbf{x}) = \beta_0 + \beta_1 \cdot \mathbf{x}_1$ with $\beta_1 \neq 0$ and $\beta_2 = 0$, i.e., the protected feature is not used by the model. In the interventional variant, the uninformativeness property implies that $\mathcal{S}(f_\beta, \mathbf{x})_2 = 0$. However, this does not mean that $Z = \mathbf{X}_2$ is independent of the output because $f_\beta(\mathbf{X}) = \beta_0 + \beta_1 \cdot \mathbf{X}_1 \not\perp \mathbf{X}_2$. In the observational variant, Aas et al. (2021) show that:

$$val(T) = \sum_{i \in N \setminus T} \beta_i \cdot E[\mathbf{X}_i | \mathbf{X}_T = \mathbf{x}_T^\star] + \sum_{i \in T} \beta_i \cdot \mathbf{x}_i^\star$$

from which, we calculate: $\mathcal{S}(f_\beta, \mathbf{x}^\star)_2 = \frac{\beta_1}{2} E[\mathbf{X}_1 | \mathbf{X}_2 = \mathbf{x}_2^\star]$. We have $\mathcal{S}(f_\beta, \mathcal{D}_\mathbf{X})_2 \equiv 0 \Leftrightarrow E[\mathbf{X}_1 | \mathbf{X}_2 = \mathbf{x}_2^\star] = 0$ for all $\mathbf{x}^\star$ in $\mathcal{D}_\mathbf{X}$. For the marginal distribution $P(\mathbf{X}_1 = v) = 1/4$ for $v = 1, -1, 2, -2$, and considering that $\mathbf{X}_2 = \mathbf{X}_1^2$, it holds that $E[\mathbf{X}_1 | \mathbf{X}_2 = v] = 0$ for all $v$. Thus $\mathcal{S}(f, \mathcal{D}_\mathbf{X})_2 \equiv 0$. However, again $f_\beta(\mathbf{X}) = \beta_0 + \beta_1 \cdot \mathbf{X}_1 \not\perp \mathbf{X}_2$.

The counterexample shows that focusing only on the Shapley values of the protected feature is not a viable way to prove DP of a model – and, a fortiori by Lemma 4.1, neither to prove ED of the model.

# E  Observational vs Interventional Shapley Values: True to the Model or True to the Data?

Many works discuss the application of Shapley values for feature attribution in ML models (Strumbelj and Kononenko, 2014; Lundberg et al., 2020; Lundberg and Lee, 2017; Lundberg et al., 2018). However, the correct way to connect a model to a coalitional game, which is the central concept of Shapley values, is a source of controversy, with two main approaches: an interventional (Aas et al., 2021; Frye et al., 2020; Zern et al., 2023), and an observational (Sundararajan and Najmi, 2020; Datta et al., 2016; Mase et al., 2019) formulation of the conditional expectation, see Eqs. (4) and (5).

In the following experiment, we compare the impact of the two approaches on our Equal Treatment Inspector to measure Explanation Disparity. We benchmark this experiment on the four prediction tasks based on the US census data (Ding et al., 2021) and use linear models for both the $f_\theta(\mathbf{X})$ and $g_\psi(\mathcal{S}(f_\theta, \mathbf{X}))$. We calculate the two variants of Shapley values using the SHAP linear explainer.[6] The comparison will be parametric to a feature perturbation hyperparameter. The interventional SHAP values break the dependence structure between features in the model to uncover how the model would behave if the inputs was changed (as it was an intervention). This option is said to stay "true to the model", meaning it will only give allocation credit to the features that the model actually uses. On the other hand, the full conditional approximation of the SHAP values respects the correlations of the input features. If the model depends on one input that is correlated with another input, then both get some credit for the model's behaviour. This option is said to say "true to the data", meaning it only considers how the model would behave when respecting the correlations in the input data (Chen et al., 2020). We measure the difference between the two approaches by looking at the AUC and the linear coefficients of the inspector $g_\psi$, for this case only for the pair of values White-Other of the "Race" attribute. Table 2 and Table 3 show that differences in AUC and coefficients are negligible.

Table 2: AUC comparison of the "Equal Treatment Inspector" to measure Explanation Disparity between estimating the Shapley values between the interventional and the observational approaches for the four prediction tasks based on the US census dataset. The % column is the relative difference.

|  | Interventional | Observational | % |
|---|---|---|---|
| Income | 0.736438 | 0.736439 | 1.1e-06 |
| Employment | 0.747923 | 0.747923 | 4.44e-07 |
| Mobility | 0.690734 | 0.690735 | 8.2e-07 |
| Travel Time | 0.790512 | 0.790512 | 3.0e-07 |

---

[6] https://shap.readthedocs.io/en/latest/generated/shap.explainers.Linear.html

Table 3: Linear regression coefficients comparison of the "Equal Treatment Inspector" between estimating the Shapley values between the interventional and the observational approaches for the ACS Income prediction task. The % column is the relative difference.

|  | Interventional | Observational | % |
|---|---|---|---|
| Marital | 0.348170 | 0.348190 | 2.0e-05 |
| Worked Hours | 0.103258 | -0.103254 | 3.5e-06 |
| Class of worker | 0.579126 | 0.579119 | 6.6e-06 |
| Sex | 0.003494 | 0.003497 | 3.4e-06 |
| Occupation | 0.195736 | 0.195744 | 8.2e-06 |
| Age | -0.018958 | -0.018954 | 4.2e-06 |
| Education | -0.006840 | -0.006840 | 5.9e-07 |
| Relationship | 0.034209 | 0.034212 | 2.5e-06 |

# F    Experiments on Datasets derived from the US Census

In the main body of the paper, we considered the ACS Income dataset. Here, we experiment with additional datasets derived from the US census database (Ding et al., 2021): ACS Travel Time, ACS Employment and ACS Mobility. We compare fairness of the prediction tasks for pairs of "Race" protected attribute groups over the California 2014 district data. For instance, for the pair White-Other, this means that $Z = 0$ represents *Race=White*, and $Z = 1$ represents *Race=Other*.

We follow the same methodology as in the experimental Section 5.2. The choice of xgboost (Chen and Guestrin, 2016) for the model $f_\theta$ is motivated as it achieves state-of-the-art performance Grinsztajn et al. (2022); Elsayed et al. (2021); Borisov et al. (2021). The choice of logistic regression for the inspector $g_\psi$ is motivated by its direct interpretability.

## F.1    ACS Employment

The goal of this task is to predict whether an individual is employed. Figure 4 shows a low DP violation, compared to the other prediction tasks based on the US census dataset. The AUC of the Equal Treatment Inspector is ranging from 0.60 to 0.75. Figure 4 (right) shows the Wasserstein distance between the coefficients of the linear regressor $g_\psi$ compared to a baseline where groups are assigned at random in the input dataset. On average, the most important features across all group comparisons are "Education" and "Citizenship".

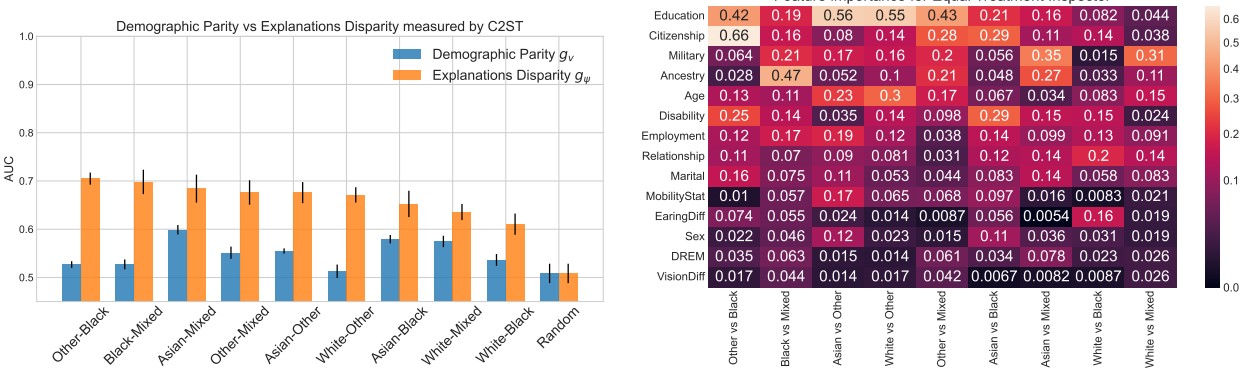

Figure 4: Left: AUC of the inspector for ET and DP, over the district of California 2014 for the ACS Employment dataset. Right: contribution of features to the ET inspector performance.

## F.2 ACS Income

The goal of this task is to predict whether an individual's income is above \$50,000,. Figure 5 shows an AUC for the ET inspector in the range of 0.60 to 0.80. By looking at the features, they highlight different ED drivers depending on the pair-wise comparison made. Figure 5 (right) shows the Wasserstein distance between the coefficients of the linear regressor $g_\psi$ compared to a baseline where groups are assigned at random in the input dataset. This feature importance post-hoc explanation method provides insights into the impact of different features as sources of unequal treatment. We observe – on average – "Education" and "Occupation" as a highly discriminatory proxy. This allows us to identify areas where adjustments or interventions may be needed to move closer to the ideal of equal treatment.

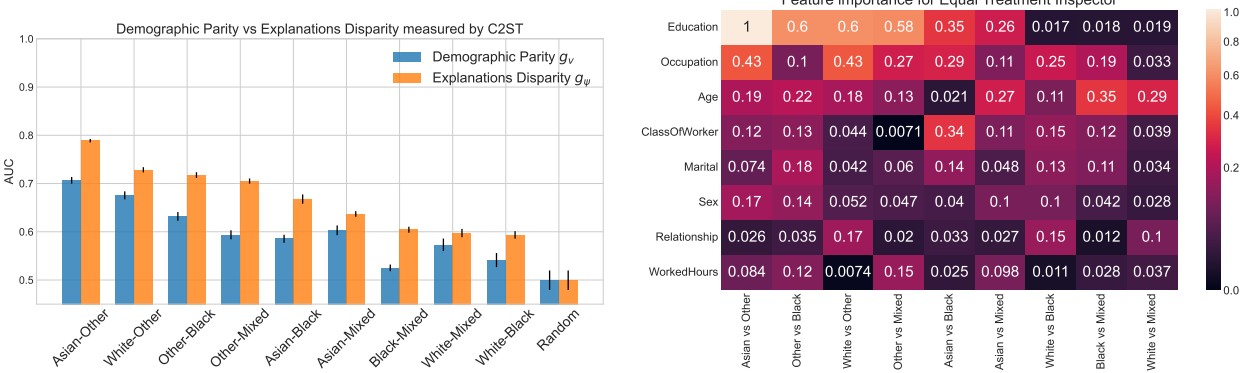

Figure 5: Left: AUC of the inspector for ED and DP, over the district of California 2014 for the ACS Income dataset. Right: contribution of features to the ED inspector performance.

## F.3 ACS Mobility

The goal of this task is to predict whether an individual had the same residential address one year ago, only including individuals between the ages of 18 and 35. This filtering increases the difficulty of the prediction task, as the base rate of staying at the same address is above 90% for the general population (Ding et al., 2021). Figure 6 show an AUC of the Equal Treatment inspector in the range of 0.60 to 0.75. By looking at the features, they highlight different source of the ED violation depending on the group pair-wise comparison. On average, the feature "Ancestry", i.e. "ancestors' lives with details like where they lived, who they lived with, and what they did for a living", plays a high relevance when predicting Explanation Disparity violation, followed by the feature "education".

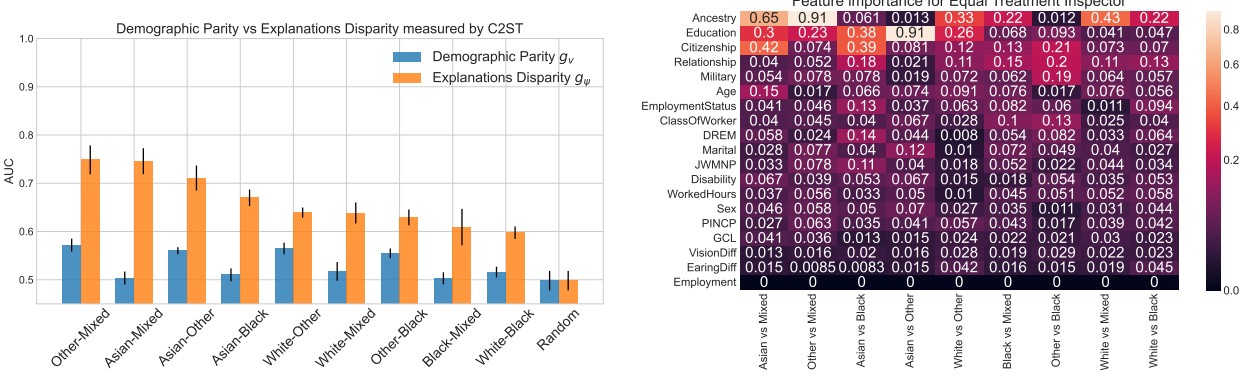

Figure 6: Left: AUC of the inspector for ED and DP, over the district of California 2014 for the ACS Mobility dataset. Right: contribution of features to the ET inspector performance.

## G   Additional Experiments

In this section, we run additional experiments on C2ST, hyperparameters, and models for estimators $f_\theta$ and inspectors $g_\psi$.

### G.1   Statistical Independence Test via Classifier AUC Test

We complement the experiments of Section 5.2 by reporting in Table 4 the results of the C2ST for group pair-wise comparisons. As discussed in Section 4.2.1, we perform the statistical test $H_0 : AUC = 1/2$ of the Equal Treatment Inspector using a Brunner-Munzel one tailed test against $H_1 : AUC > 1/2$ as implemented by Virtanen et al. (2020). Table 4 reports the empirical AUC on the test set, the confidence intervals at 95% confidence level (columns "Low" and "High"), and the p-value of the test. The "Random" row regards a randomly assigned group and represents a baseline for comparison. The statistical tests clearly show that the AUC is significantly different from $1/2$, also when correcting for multiple comparison tests.

Table 4: Results of the C2ST on the Equal Treatment Inspector.

| *Pair* | AUC | Low | High | pvalue | Test Statistic |
|---|---|---|---|---|---|
| *Random* | 0.501 | 0.494 | 0.507 | 0.813 | 0.236 |
| *White-Other* | 0.735 | 0.731 | 0.739 | $< 2.2e\text{-}16$ | 97.342 |
| *White-Black* | 0.62 | 0.612 | 0.627 | $< 2.2e\text{-}16$ | 27.581 |
| *White-Mixed* | 0.615 | 0.607 | 0.624 | $< 2.2e\text{-}16$ | 23.978 |
| *Asian-Other* | 0.795 | 0.79 | 0.8 | $< 2.2e\text{-}16$ | 107.784 |
| *Asian-Black* | 0.667 | 0.659 | 0.676 | $< 2.2e\text{-}16$ | 38.848 |
| *Asian-Mixed* | 0.644 | 0.634 | 0.653 | $< 2.2e\text{-}16$ | 28.235 |
| *Other-Black* | 0.717 | 0.708 | 0.725 | $< 2.2e\text{-}16$ | 48.967 |
| *Other-Mixed* | 0.697 | 0.688 | 0.707 | $< 2.2e\text{-}16$ | 39.925 |
| *Black-Mixed* | 0.598 | 0.586 | 0.61 | $< 2.2e\text{-}16$ | 15.451 |

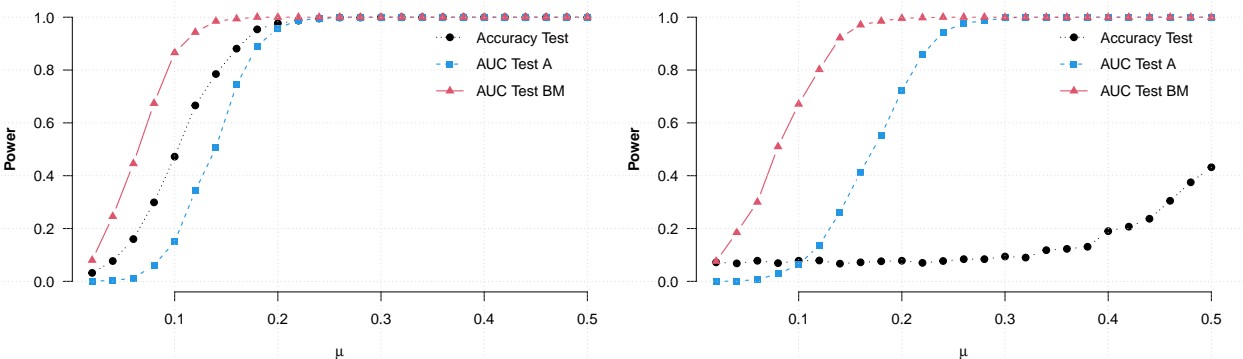

Figure 7: Comparing the power (the higher, the better) of C2ST based on AUC with Brunner-Munzel test (AUC Test BM) vs Accuracy vs AUC with asymptotic normal approximation of the Wilcoxon–Mann–Whitney statistics (AUC Test A). Left: balanced groups ($P(Z = 1) = 0.5$). Right: unbalanced groups ($P(Z = 1) = 0.2$).

We also compare the power of the C2ST based on the AUC using the Brunner-Munzel test against the two-sample test of Lopez-Paz and Oquab (2017), which is based on accuracy, and against the AUC test of Chakravarti et al. (2023), which is based on the asymptotic normal approximation of the Wilcoxon–Mann–Whitney statistics. We generate synthetic datasets from $\mathbf{X} \times Z$, where $Z \sim Ber(0.5)$ (balanced groups) or $Z \sim Ber(0.2)$ (unbalanced groups), and $\mathbf{X} = (\mathbf{X}_1, \mathbf{X}_2)$ with positives ($Z = 1$) distributed as $N(\begin{bmatrix} \mu & \mu \end{bmatrix}, \mathbf{\Sigma})$ and negatives ($Z = 0$) distributed as $N(\begin{bmatrix} -\mu & -\mu \end{bmatrix}, \mathbf{\Sigma})$, where $\mathbf{\Sigma} = \begin{bmatrix} 1 & 0.5 \\ 0.5 & 1 \end{bmatrix}$. Thus, the larger the parameter $\mu$, the easier is to distinguish the distributions of positives and negatives. Figure 7

reports the power (i.e., the probability of rejecting $H_0$ when it does not hold) of the three tests using in all cases a logistic regression classifier. The power is estimated by 1,000 runs for each of the $\mu$'s ranging from 0.005 to 0.5. The figure highlights that, under such a setting, testing the AUC using the Brunner-Munzel test achieves a better power than using accuracy or using an asymptotic test. Our approach exhibits the best power, and the difference is higher in the case groups are unbalanced.

## G.2 Hyperparameters Evaluation

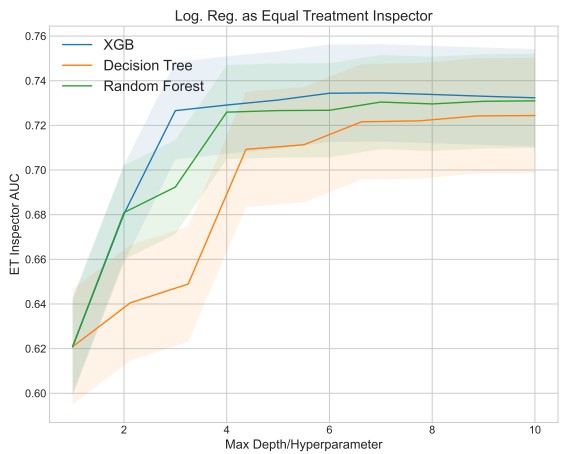
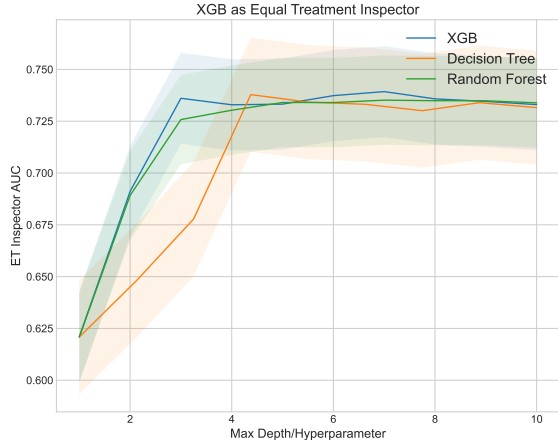

Figure 8: AUC of the Equal Treatment Inspector for Explanation Disparity, over the district of CA14 for the US Income dataset.

This section extends our experimental setup, where we increase the model complexity by varying the model hyperparameters. We use the US Income dataset for the population of the CA14 district. We consider three models for $f_\theta$: Decision Trees, Gradient Boosting, and Random Forest. For the Decision Tree models, we vary the depth of the tree, while for the Gradient Boosting and Random Forest models, we vary the number of estimators. Shapley values are calculated by the TreeExplainer algorithm (Lundberg et al., 2020). For the ET inspector $g_\psi$, we consider logistic regession, and XGB.

Figure 8 shows that less complex models, such as Decision Trees with maximum depth 1 or 2, are also less unfair. However, as we increase the model complexity, the unequal treatment of the model becomes more pronounced, achieving a plateau when the model has enough complexity. Furthermore, when we compare the results for different inspectors, we observe minimal differences (note that the y-axis takes different ranges). In our experiments of the main body, we used XGBoost due to its state-of-the-art performance.

## G.3 Varying Estimator and Inspector

We vary here the model $f_\theta$ and the inspector $g_\psi$ over a wide range of well-known classification algorithms. Table 5 shows that the choice of model and inspector impacts on the measure of Equal Treatment, namely the AUC of the inspector. By Theorem 4.3, the larger the AUC of any inspector the smaller is the $p$-value of the null hypothesis $\mathcal{S}(f_\theta, \mathbf{X}) \perp Z$. Therefore, inspectors able to achieve the best AUC (higher) should be considered. Weak inspectors have a lower power, i.e., a lower probability of rejecting the null hypothesis when it does not hold.

## G.4 Experiment: Explaining Un-Equal Treament

We complement the results of the experimental Section 5.1 with a further experiment relating the correlation hyperparameter $\gamma$ to the coefficients of an explainable ET inspector. We consider a synthetic dataset with one more feature, by drawing $10,000$ samples from a $\mathbf{X}_1 \sim N(0,1)$, $\mathbf{X}_2 \sim N(0,1)$, and $(\mathbf{X}_3, \mathbf{X}_5)$ and $(\mathbf{X}_4, \mathbf{X}_5)$ following bivariate normal distributions $N\left(\begin{bmatrix} 0 & 0 \end{bmatrix}, \begin{bmatrix} 1 & \gamma \\ \gamma & 1 \end{bmatrix}\right)$ and $N\left(\begin{bmatrix} 0 & 0 \end{bmatrix}, \begin{bmatrix} 1 & \gamma/2 \\ \gamma/2 & 1 \end{bmatrix}\right)$, respectively. We

Table 5: AUC of the Equal Treatment inspector for different combinations of models and inspectors.

| Inspector $g_\psi$ | Model $f_\theta$ | | | | |
|---|---|---|---|---|---|
| | DecisionTree | SVC | Logistic Reg. | RF | XGB |
| DecisionTree | 0.631 | 0.644 | 0.644 | 0.664 | 0.634 |
| KNN | 0.737 | 0.754 | 0.75 | 0.744 | 0.751 |
| Logistic Reg. | 0.767 | 0.812 | 0.812 | 0.812 | 0.821 |
| MLP | 0.786 | 0.795 | 0.795 | 0.813 | 0.804 |
| RF | 0.776 | 0.782 | 0.781 | 0.758 | 0.795 |
| SVC | 0.743 | 0.807 | 0.807 | 0.790 | 0.810 |
| XGB | 0.775 | 0.780 | 0.780 | 0.789 | 0.790 |

define the binary protected feature $Z$ with values $Z = 1$ if $\mathbf{X}_5 > 0$ and $Z = 0$ otherwise. As in Section 5.1, we consider two experimental scenarios. In the first scenario, the indirect case, we have unfairness in the data and in the model. The targe feature is $Y = \sigma(\mathbf{X}_1 + \mathbf{X}_2 + \mathbf{X}_3 + \mathbf{X}_4)$, where $\sigma$ is the logistic function. In the second scenario, the uninformative case, we have unfairness in the data and fairness in the model. The target feature is $Y = \sigma(\mathbf{X}_1 + \mathbf{X}_2)$.

Figure 9 shows how the coefficients of the inspector $g_\psi$ vary with correlation $\gamma$ in both scenarios. In the indirect case, coefficients for $\mathcal{S}(f_\theta, \mathbf{X})_1$ and $\mathcal{S}(f_\theta, \mathbf{X})_2$ correctly attributes zero importance to such variables, while coefficients for $\mathcal{S}(f_\theta, \mathbf{X})_3$ and $\mathcal{S}(f_\theta, \mathbf{X})_4$ grow linearly with $\gamma$, and with the one for $\mathcal{S}(f_\theta, \mathbf{X})_3$ with higher slope as expected. In the uninformative case, coefficients are correctly zero for all variables.

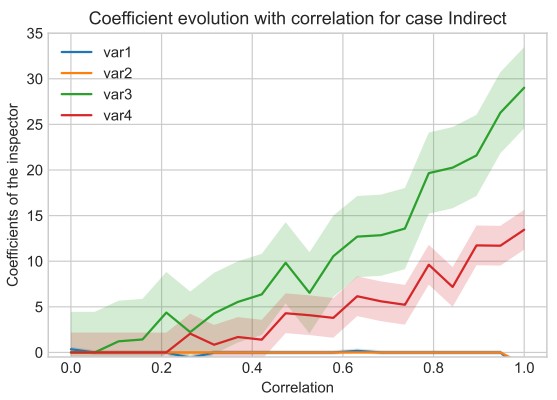
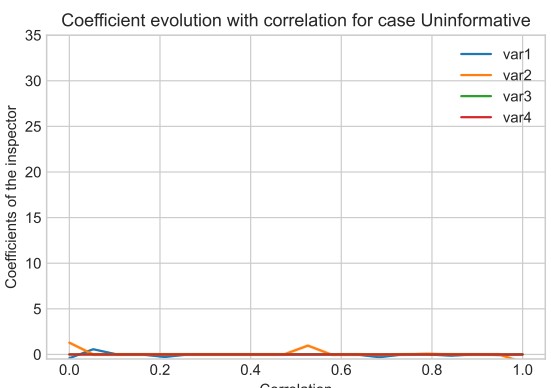

Figure 9: Coefficient of $g_\psi$ over $\gamma$ for synthetic datasets in two experimental scenarios.

### G.5 Statistical Comparison of Demographic Parity versus Explanation Disparity

So far, we measured ED and DP fairness using the AUC of an inspector, $g_\psi$ and $g_v$, respectively (see Section 5). For DP, however, other probability distance metrics can be considered, including the p-value of the Kolmogorov–Smirnov (KS) test and the Wasserstein distance. Table 6 reports all such distances in the format "mean $\pm$ stdev" calculated over 100 random sampled datasets. The pairs of group comparisons are sorted by descending AUC values. We highlight in red the values below the mean threshold of 0.05 for the KS test (above no violation), 0.55 for the AUC of the C2ST (below no violation), and 0.05 for the Wasserstein distance (below no violation). They represent cases where ED violation occurs, but no DP violation is measured (with different metrics).

Table 6: Comparison of ED and DP measured in different ways. Cases of Explanation Disparity violations but no Demographic Parity violations are highlighted in red.

| Pair | Data | Explanation Disparity | Demographic Parity | | |
|---|---|---|---|---|---|
| | | C2ST (AUC) | C2ST (AUC) | KS (pvalue) | Wasserstein |
| Asian-Other | Income | $0.794 \pm 0.004$ | $0.709 \pm 0.004$ | $0.338 \pm 0.007$ | $0.256 \pm 0.004$ |
| White-Other | Income | $0.734 \pm 0.002$ | $0.675 \pm 0.003$ | $0.282 \pm 0.003$ | $0.209 \pm 0.002$ |
| Other-Black | Income | $0.724 \pm 0.004$ | $0.628 \pm 0.006$ | $0.216 \pm 0.007$ | $0.143 \pm 0.004$ |
| Other-Mixed | Income | $0.707 \pm 0.005$ | $0.593 \pm 0.005$ | $0.169 \pm 0.006$ | $0.117 \pm 0.004$ |
| Asian-Black | Income | $0.664 \pm 0.008$ | $0.587 \pm 0.004$ | $0.142 \pm 0.005$ | $0.111 \pm 0.004$ |
| Asian-Mixed | Income | $0.644 \pm 0.005$ | $0.607 \pm 0.006$ | $0.159 \pm 0.008$ | $0.128 \pm 0.006$ |
| White-Mixed | Income | $0.613 \pm 0.005$ | $0.546 \pm 0.005$ | $0.082 \pm 0.004$ | $0.058 \pm 0.002$ |
| White-Black | Income | $0.613 \pm 0.005$ | $0.57 \pm 0.007$ | $0.113 \pm 0.008$ | $0.080 \pm 0.006$ |
| Black-Mixed | Income | $0.603 \pm 0.006$ | $0.523 \pm 0.007$ | $0.055 \pm 0.007$ | $0.023 \pm 0.004$ |
| Asian-Black | TravelTime | $0.677 \pm 0.052$ | $0.502 \pm 0.011$ | $0.021 \pm 0.009$ | $0.01 \pm 0.003$ |
| Asian-Other | TravelTime | $0.653 \pm 0.024$ | $0.528 \pm 0.006$ | $0.053 \pm 0.011$ | $0.027 \pm 0.004$ |
| Asian-Mixed | TravelTime | $0.647 \pm 0.013$ | $0.557 \pm 0.003$ | $0.096 \pm 0.004$ | $0.045 \pm 0.002$ |
| White-Other | TravelTime | $0.636 \pm 0.020$ | $0.568 \pm 0.007$ | $0.107 \pm 0.010$ | $0.060 \pm 0.005$ |
| Other-Mixed | TravelTime | $0.618 \pm 0.017$ | $0.546 \pm 0.011$ | $0.079 \pm 0.012$ | $0.043 \pm 0.006$ |
| Other-Black | TravelTime | $0.615 \pm 0.021$ | $0.526 \pm 0.011$ | $0.049 \pm 0.014$ | $0.026 \pm 0.006$ |
| White-Black | TravelTime | $0.599 \pm 0.006$ | $0.569 \pm 0.004$ | $0.120 \pm 0.006$ | $0.057 \pm 0.003$ |
| Black-Mixed | TravelTime | $0.588 \pm 0.012$ | $0.557 \pm 0.012$ | $0.098 \pm 0.015$ | $0.056 \pm 0.001$ |
| White-Mixed | TravelTime | $0.557 \pm 0.008$ | $0.497 \pm 0.006$ | $0.016 \pm 0.004$ | $0.006 \pm 0.002$ |
| Other-Black | Employment | $0.744 \pm 0.008$ | $0.524 \pm 0.005$ | $0.036 \pm 0.005$ | $0.036 \pm 0.004$ |
| Asian-Other | Employment | $0.711 \pm 0.011$ | $0.557 \pm 0.003$ | $0.066 \pm 0.004$ | $0.066 \pm 0.003$ |
| White-Other | Employment | $0.695 \pm 0.007$ | $0.524 \pm 0.003$ | $0.019 \pm 0.005$ | $0.019 \pm 0.002$ |
| Other-Mixed | Employment | $0.683 \pm 0.022$ | $0.557 \pm 0.008$ | $0.083 \pm 0.005$ | $0.083 \pm 0.003$ |
| Black-Mixed | Employment | $0.678 \pm 0.028$ | $0.534 \pm 0.007$ | $0.049 \pm 0.007$ | $0.048 \pm 0.004$ |
| Asian-Mixed | Employment | $0.671 \pm 0.019$ | $0.610 \pm 0.006$ | $0.014 \pm 0.006$ | $0.145 \pm 0.004$ |
| Asian-Black | Employment | $0.655 \pm 0.021$ | $0.587 \pm 0.004$ | $0.106 \pm 0.006$ | $0.106 \pm 0.004$ |
| White-Mixed | Employment | $0.651 \pm 0.009$ | $0.581 \pm 0.006$ | $0.095 \pm 0.004$ | $0.095 \pm 0.003$ |
| White-Black | Employment | $0.619 \pm 0.011$ | $0.544 \pm 0.004$ | $0.049 \pm 0.003$ | $0.049 \pm 0.002$ |
| Asian-Mixed | Mobility | $0.753 \pm 0.020$ | $0.511 \pm 0.014$ | $0.04 \pm 0.012$ | $0.014 \pm 0.006$ |
| Other-Mixed | Mobility | $0.748 \pm 0.020$ | $0.573 \pm 0.015$ | $0.113 \pm 0.017$ | $0.062 \pm 0.009$ |
| Asian-Other | Mobility | $0.714 \pm 0.011$ | $0.565 \pm 0.01$ | $0.114 \pm 0.011$ | $0.054 \pm 0.005$ |
| Asian-Black | Mobility | $0.672 \pm 0.012$ | $0.503 \pm 0.014$ | $0.032 \pm 0.011$ | $0.012 \pm 0.004$ |
| Other-Black | Mobility | $0.660 \pm 0.012$ | $0.526 \pm 0.009$ | $0.044 \pm 0.009$ | $0.02 \pm 0.004$ |
| White-Mixed | Mobility | $0.655 \pm 0.007$ | $0.568 \pm 0.005$ | $0.105 \pm 0.007$ | $0.044 \pm 0.003$ |
| White-Other | Mobility | $0.626 \pm 0.017$ | $0.555 \pm 0.009$ | $0.091 \pm 0.010$ | $0.046 \pm 0.005$ |
| White-Black | Mobility | $0.611 \pm 0.009$ | $0.518 \pm 0.008$ | $0.043 \pm 0.008$ | $0.017 \pm 0.004$ |
| Black-Mixed | Mobility | $0.602 \pm 0.035$ | $0.503 \pm 0.016$ | $0.031 \pm 0.013$ | $0.012 \pm 0.006$ |

## H    LIME as an Alternative to Shapley Values

Def. 3.2 of ED is parametric in the explanation function. We used Shapley values for their theoretical advantages (see Appendix B). Another widely used feature attribution technique is LIME (Local Interpretable Model-Agnostic Explanations). The intuition behind LIME is to create a local linear model that approximates the behavior of the original model in a small neighbourhood of the instance to explain (Ribeiro et al., 2016a;b), whose mathematical intuition is very similar to the Taylor-Maclaurin series. This section discusses the differences in our approach when adopting LIME instead of the SHAP implementation of Shapley values. First of all, LIME has certain drawbacks:

- **Computationally Expensive:** Its current implementation is more computationally expensive than current SHAP implementations such as TreeSHAP (Lundberg et al., 2020), Data SHAP (Kwon et al., 2021; Ghorbani and Zou, 2019), or Local and Connected SHAP (Chen et al., 2019b). This problem is exacerbated when producing explanations for multiple instances (as in our case). In fact, LIME requires sampling data and fitting a linear model, which is a computationally more expensive approach than the aforementioned model-specific approaches to SHAP. A comparison of the runtime is reported in the next sub-section.

- **Local Neighborhood:** The randomness in the calculation of local neighbourhoods can lead to instability of the LIME explanations. Works including Slack et al. (2020); Alvarez-Melis and Jaakkola (2018); Adebayo et al. (2018) highlight that several types of feature attributions explanations, including LIME, can vary greatly in their provided explanations.

- **Dimensionality:** LIME requires, as a hyperparameter, the number of features to use for the local linear model. For our method, all features in the explanation distribution should be used. However, linear models suffer from the curse of dimensionality. In our experiments, this is not apparent, since our synthetic and real datasets are low-dimensional.

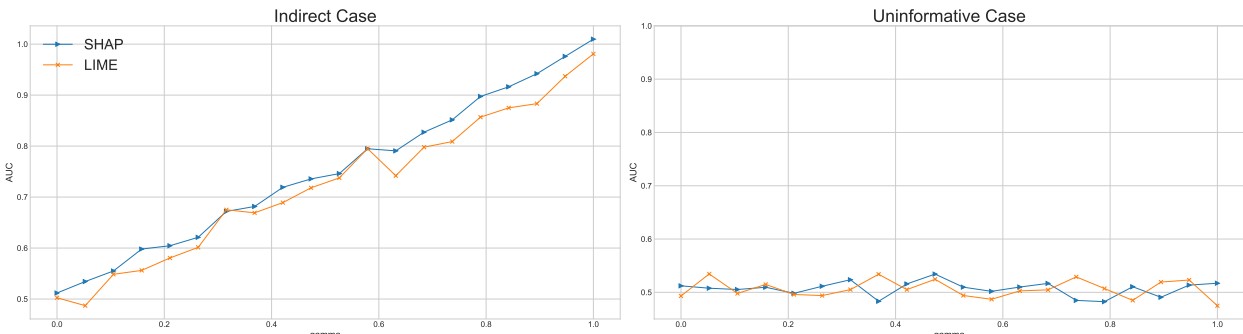

Figure 10: AUC of the Equal Treatment Inspector using SHAP vs using LIME.

Figure 10 compares the AUC of the Equal Treatment Inspector using SHAP and LIME as explanation functions over the synthetic dataset of Section 5.1. In both scenarios (indirect case and uninformative case), the two approaches have similar results. In both cases, however, the stability of using SHAP is better than using LIME.

### H.1    Runtime

We conduct an analysis of the runtimes for generating the explanation distributions using TreeShap vs LIME. We adopt `shap` version 0.41.0 and `lime` version 0.2.0.1 as software packages. In order to define the local neighborhood for both methods in this example, we used all the data provided as background data. The model $f_\theta$ is set to `xgboost`. As data we produce a randon generated data matrix, of varying dimensions. When varying the number of samples, we use 5 features, and when varying the number of features, we

use 1000 samples. Figure 11 shows the elapsed time for generating explanation distributions with varying numbers of samples and columns. The runtime required to generate explanation distributions using LIME is considerably greater than using SHAP. The difference becomes more pronounced as the number of samples and features increases.

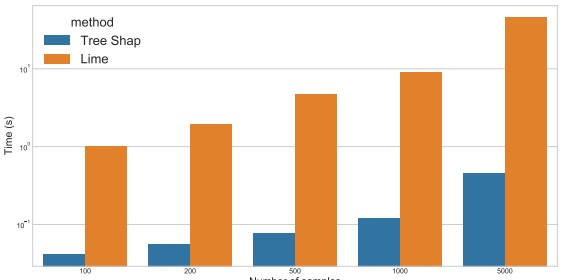 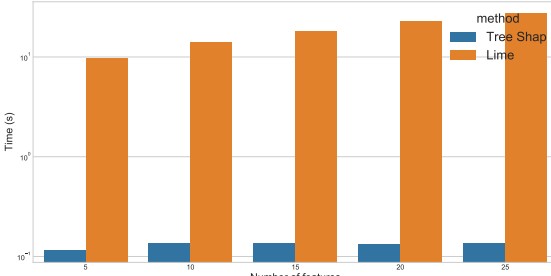

Figure 11: Elapsed time for generating explanation distributions using SHAP and LIME with different numbers of samples (left) and features (right) on synthetically generated datasets. Note that the y-scale is logarithmic.

# I  Impact Statements

## I.1  Reproducibility Statement

To ensure the reproducibility of our results, we make source code publicly available at `https://anonymous.4open.science/r/xAIAuditing-F6F9/README.md`. The Python package `explanationspace`, anonymously available at `https://anonymous.4open.science/r/explanationspace-B4B1/README.md`, will be released as open-source. We use default `scikit-learn` parameters (Pedregosa et al., 2011), unless stated otherwise. Our experiments ran on a 4 vCPU server with 32 GB RAM.

## I.2  Research Positionality Statement

Our backgrounds and experiences coming from Western education significantly influenced the trajectory of this work. As interdisciplinary researchers, our diverse perspectives enriched the examination of the notion of *equal treatment* in distributive justice in the context of AI fairness. Drawing from interdisciplinary backgrounds, including philosophy and computer science, we integrated moral theories into technical domains. Acknowledging our own positionalities, we recognize the importance of context-specific interpretations of political philosophy in different socio-cultural settings.

## I.3  Adverse Impact Statement

While we believe our exploration of liberalism-oriented politics for fairness metrics in AI contributes valuable insights, we acknowledge the potential adverse impacts of our work. One unintended consequence may be the oversimplification of philosophical considerations in AI systems. In social sciences, a longstanding critique argues that when systematic differences exist between groups, applying equal treatment may perpetuate discrimination by not providing equal opportunities to all individuals. We leave the normative discussion of which political framework or philosophical paradigm should be pursued by policy to the discourse in the social sciences and the broad public. We caution against the uncritical adoption of our proposed framework, urging ongoing dialogue and adaptation to ensure that AI ethics remains responsive to emerging challenges and ethical considerations.

