# OpenReview forum: "Measuring the Impact of Equal Treatment as Blindness via Distributions of Explanations Disparities"
_TMLR — Rejected by TMLR_

### Review · Reviewer_RgW9 · 2025-03-21

**Summary Of Contributions:**

The paper introduces a novel fairness metric called Explanation Disparity (ED) that assesses whether machine learning models treat individuals equally by examining their explanation distributions—specifically through Shapley values. They propose the Equal Treatment Inspector, a framework that leverages a classifier two-sample test (based on AUC) to test whether the explanation distribution is independent of protected attributes.

**Audience:**

Yes

**Broader Impact Concerns:**

No broader impact concerns.

**Claims And Evidence:**

Yes

**Requested Changes:**

I think the work could be strengthened by implementing an explainability method other than Shapley values into the framework. Shapley values are quite computationally expensive, so I think showing the flexibility of the method would add significant value.

**Strengths And Weaknesses:**

**Strengths**:
- **Innovative Fairness Metric**: The paper proposes Explanation Disparity, a new metric that looks deeper than traditional Demographic Parity by examining how decisions are made (via explanation distributions) rather than just the final outputs.

- **Experimental Validation**: The authors validate their approach with both synthetic and real-world datasets, demonstrating that their method can detect fairness issues that conventional metrics might miss.

- **Practical Contributions**: The release of an open-source package enhances reproducibility and gives practitioners tools to audit fairness in their models.

**Weaknesses**:
- **Computational Considerations**: Shapley value approximations, while theoretically robust, can be computationally intensive and may introduce errors in practical applications.

- **Assumptions and Real-World Complexity**: Some theoretical assumptions may not hold in more complex, real-world scenarios where data distributions and correlations are more intricate.

---

### Review · Reviewer_rCem · 2025-04-27

**Summary Of Contributions:**

The paper proposes *explanation disparity* as an alternative for demographic parity evaluation in *fairness as unawareness* setting. The authors propose to use Shapley value distributions across groups as a proxy for fairness, by training an inspector model that predicts the protected attribute. The inspector model can also help highlight attributes contributing to such disparity.

**Audience:**

Yes

**Broader Impact Concerns:**

The authors should address the reliability of the explanation mentioned above.

**Claims And Evidence:**

No

**Requested Changes:**

- "we show that Demographic Parity may indicate fairness, even when groups are being treated differently by the model": clarify how this differs from notions such as "fairness gerrymandering" (Kearns et al., 2018)

- Make the assumption on explanation reliability clear in the paper, or explain how this limitation can be addressed in real-world settings. I suggest "Deck et al., 2023" as a relevant literature to consider for this discussion.

- Clarify if "explanation disparity" and "demographic parity" need the same query budget for a reliable estimation.

- Report and discuss the results on equal opportunity and individual fairness.


(Deck et al., 2023):  A Critical Survey on Fairness Benefits of XAI

**Strengths And Weaknesses:**

## Strengths
- The authors propose an open-source Python package to facilitate the use of the proposed workflow for unfairness quantification and attribution.

- The authors evaluate on both synthetic and real-world data and perform extensive experiments.

## Weaknesses

- The main weakness of the contribution is the implicit assumption that the explanations used as input for C2ST are reliable. This is a strong assumption, as several works have demonstrated these explanations can be unreliable, even in non-adversarial settings.

- Demographic parity (as an independence criterion) has already been extensively criticized in the fairness literature and motivates the separation criterion (cf. Barocas et al., 2023). This makes demographic parity already a weaker baseline. So, comparing the proposed metric with other metrics would have been more relevant. From Table 1, individual fairness seems like a good candidate. Also, EO is defined but not used in evaluation.

(Barocas et al., 2023): Fairness and Machine Learning: Limitations and Opportunities

---

> ### Author Response · Authors · 2025-05-04
>
> Dear reviewer,
>
> Many thanks for your comments!
> i) We agree that our method depends on the quality and reliability of model explanations. While improving explanation reliability is not the focus of this work, we acknowledge its importance. Our software implementation supports explainers such as SHAP and LIME, and future updates to these libraries can be readily integrated into our open-source package. Thanks for the suggestion of Deck et al., 2023, which we will incorporate along the Impact Statement and Limitations section.
>
> ii) Thank you for pointing out the connection to fairness gerrymandering (Kearns et al., 2018). We will clarify the distinction in the revised manuscript. In particular, their definition of 2.1, relies on group-specific error rates (e.g., false positives), which require access to ground truth labels. In many real-world applications, such as credit lending, obtaining outcome labels can be delayed by years, making our method more broadly applicable in settings where labels are scarce, biased or delayed.
>
> iii) Equal Opportunity is discussed on page 19 and Individual Fairness on page 21 of the current version. We would be happy to expand these discussions if more detailed analysis or clarification is required.
>
> iv) Regarding query budget: Appendix H.1 includes an experiment comparing computational requirements. Explanation disparity does require more computation than demographic parity due to the need to generate feature attributions. However, this remains tractable—our experiments were conducted on a standard laptop and completed in minutes (see Figure 11). We will make this trade-off clearer in the main text.
>
> Once again, thank you for your helpful comments. We will incorporate the suggested changes to strengthen the manuscript.

---

> > ### Author Response · Authors · 2025-05-13
> >
> > > I think it's a stretch to describe the C2ST they use as novel. Using Brunner-Munzel instead of some other test for the null that the AUC is 0.5 is, in my opinion, a reasonable but very minor variation.
> >
> > We comment at the end of Section 4.2.1 that the Brunner-Munzel is preferable over other choices: the Wilcoxon–Mann–Whitney test assumes same-shape distribution between positives and negatives; the Fligner–Policello assumes symmetric distributions. We agree that the choice of a specific test is a minor variation, but, among all variations, it is important to choose the one that works best. See also next answer.
> >
> > > The characterization of the C2ST test used by Chakravarti et al. (2023) as relying on an asymptotic normal distribution seems to be slightly inaccurate; Chakravarti et al. mention using three different methods to derive the null distribution: the asymptotic distribution, bootstrap, and permutation methods. It's not clear to me that Brunner-Munzel would outperform this test with the bootstrap or permutation methods.
> >
> > Thanks for this comment. Following your question, we implemented the bootstrap and permutation variants. Preliminary experiments show that the power of such variants is comparable to the Brunner-Munzel test, which however is faster (unsurprisingly). Since the statistical test is not a key point (see previous answer), and actually it can be a parameter method of the approach, we will claim that the state-of-the-art method should be used, and that the default method used in our implementation is the Brunner-Munzel test based on both performance and computational efficiency. Appendix G.1 will be expanded reporting experiments with the bootstrap and permutation methods. Also, we will experiment with additional methods we recently were aware of, ed in particular: https://doi.org/10.1002/bimj.201500105. The main objective of Appendix G.1 is to discuss possible variants of the test method.
> >
> > > Furthermore, the empirical comparison of Brunner-Munzel to the Mann-Whitney U-based tests looked at power but did not look at Type 1 error rates, so it's somewhat limited.
> >
> > Figure 7 in Appendix G.1 reports power at variation of $\mu$ ranging from 0.02 to 0.50 at significance level $\alpha=0.05$. Notice that for $\mu$ equal to 0.00 the plot would show the Type 1 error. While this is not shown in the figure, the trend of the lines is clear: Type 1 error of Brunner-Munzel is is close to the expected $\alpha=0.05$, while Type 1 error of Wilcoxon–Mann–Whitney approaches $0$ (and the same for the accuracy test). In summary, Wilcoxon–Mann–Whitney is less likely to detect real dependence, which in our context, means it is less likely to detect disparity. However, considering the bootstrap and permutation variants by Chakravarti et al. (2023), their Type 1 error is comparable to the one of the Brunner-Munzel. We will expand the discussion in Appendix G1 to include Type 1 error of the various methods.

---

> > ### Comment · Reviewer_rCem · 2025-05-31
> > **Thank you for the response**
> >
> > Thank you for your clarification of (ii) and (iii). Please find below more clarifications on (i) and (iv):
> >
> > (i) Besides citing work that discusses the reliability of explanations, I believe it would be valuable to conduct an experiment demonstrating how explanation sensitivity affects test results. For instance, investigating potential arbitrariness in predicting the sensitive attribute for the same instance across multiple runs of the explanation method. Additionally, the authors should clarify who the intended explainee is. Assuming reliability is reasonably addressed, this method appears more suited to model developers than to auditors or affected parties, since explanations can be manipulated (cf. adversarial XAI: https://github.com/hbaniecki/adversarial-explainable-ai).
> >
> >
> > (iv) I was thinking more like an analysis as in Fig. 2, showing performance of *Explanation Disparity* for several value of the size of $D_X^{val}$

---

> > > ### Author Response · Authors · 2025-06-01
> > >
> > > (i) That’s an interesting suggestion. For instance, estimating "uncertainty values" could help illustrate the robustness or potential arbitrariness of the method. While this could broaden the scope of the paper by adding a statistical perspective on reliability, conducting a thorough analysis may require several additional experiments, possibly exceeding the intended scope of this work. However, if the reviewer deems it necessary, we are open to expanding the appendix to include a preliminary experiment in this direction.
> > >
> > > Regarding the first explanation distributions (the first blue box in Fig. 1), this part does not involve human-computer interaction and therefore does not require an "explainee" in the usual sense. In contrast, the second blue box of Fig 1 might involve such interaction. In this work, the intended "persona" is a technically skilled individual capable of understanding the proposed method’s underlying mechanisms which could be developers/auditors.
> > >
> > > (iv) Since we are applying a binary classifier, experiments where one class is underrepresented fall within the domain of imbalanced learning in machine learning. If the reviewer finds it appropriate, we can include an additional experiment in the appendix to demonstrate how the ED Inspector’s performance varies with different dataset sizes, even thought this a well explored area of ML.
> > >
> > > [one of the responses of this thread should have been in the other rebuttal. We will change it.]

---

### Review · Reviewer_XmXv · 2025-04-28

**Summary Of Contributions:**

The authors introduce a fairness criterion called Explanation Disparity (ED), which requires statistical independence between a sensitive feature and the Shapley values for a model's input features. They show theoretically that if ED holds, then Demographic Parity (DP) holds, while the converse is not true in general. They use a variant of a classifier two-sample test (C2ST) to test for violations of ED. They illustrate their definition and method on synthetic and real datasets.

**Audience:**

Yes

**Broader Impact Concerns:**

No concerns.

**Claims And Evidence:**

No

**Requested Changes:**

While I don't think machine learning researchers have to come up with fully fleshed out ethical justifications for fairness measures, I find the motivation for this work very confusing, and as a result I find it hard to assess the potential impact of this work. I do find the notion of Explanation (Dis)parity intuitively appealing, but I'm struggling to understand whether that's an aesthetic appeal or whether this really addresses some problem in the field. I'm hopeful that the authors can frame the motivation and the evidence more clearly to answer this question.

**Strengths And Weaknesses:**

Strengths
---------
The use of the distribution of Shapley values is an interesting idea that has intuitive appeal, insofar as Shapley values capture how feature values affect model outputs. The mathematical formulation is clear and precise, and the experimental setup is generally clear. The use of a classifier two-sample test to test for violations of ED makes sense. The overflow workflow presented in Figure 1 is clear.


Weaknesses
----------
The authors ground their fairness definition with reference to fairness through unawareness, and more generally to philosophical notions of equal treatment. I find the framing and motivation quite muddled, however. For example, the authors claim that fairness through unawareness (FTU) can "be measured using disparate treatment metrics such as Demographic Parity". FTU is conventionally understood to be satisfied precisely when a model doesn't use a sensitive feature as one of its inputs, so I don't really know what is meant by "measuring" FTU. (It could be the case that one would want to estimate whether FTU holds in a setting where researchers have access to model outputs only and not model inputs, but this is not the problem setting described in this paper.) They say "Demographic Parity (DP), also called Statistical Parity, which compares the distributions of predicted outcomes of a model $f$ for different social groups has been proposed as a potential metric to measure this effect by researchers." However, the papers they cite do not seem to support this claim. For example, my reading of Simons et al. (2021) is that it is precisely about the tension between equal treatment (e.g. FTU) and equality of outcomes (e.g. DP). I would say it is well understood in the literature that FTU does not entail DP or Equalized Odds or any other fairness definition that depends on the distribution of the model inputs or outputs, so this seems like a strawman argument.

The authors motivate their fairness definition as being designed to "measure the longstanding criticism of" FTU. Given that FTU doesn't entail other fairness definitions, I would say that any other fairness definition can be used to critique FTU. I'm therefore unsure what their fairness definition in particular offers. In Section 2.3, the authors give six criteria, (R0)-(R5) that motivate their definition, but they're unclear to me. For example, (R1) doesn't have a main verb. In (R3), it's not clear what counts as background knowledge. (R4) mentions biased decisions, but the the whole enterprise here is to define what constitutes a biased decision, so I'm not really sure what that means. (R5) seems orthogonal to fairness measures themselves, since explanations are generally derived separately. This is true of their method as well; in Figure 1, the "Explain Equal Treatment Inspector" step is separate from the step of determining whether unequal treatment exists. As a result of all of this, I'm not really sure how to read Table 1.

On a practical front, it's not clear to me if ED solves a problem that DP doesn't. In Figure 2, for example, while the ED-based methods have a steeper slope than the DP-based method, it's not clear whether this difference is meaningful. The point here is to detect whether a disparity exists, so the real question is whether the DP-based method correctly or incorrectly fails to reject the null that the AUC is 0.5 at different rates than the ED-based methods. Adding confidence intervals would help, or maybe plotting the power of the test rather than the AUC against $\gamma$ would be more compelling.

On the Time Travel Data, the fact that the ED-based and DP-based methods yields different results doesn't imply that "there are instances where DP fails to identify discrimination that ED successfully detects." There's no ground truth about what constitutes discrimination here; DP and ED are themselves two different definitions of discrimination, so it's not surprising that the corresponding measures would be different.

Overall, it's unclear to me what problem their fairness definition addresses, or what philosophical or ethical notion it instantiates.

There are a couple other issues:
- Why are they using a C2ST to measure violations of demographic parity? This seems unrelated to the unfairness measure given right below Definition 2.1.
- Why is equalized odds introduced if it's never measured?
- The measure of unfairness in the input should be moved up with Definitions 2.1 and 2.2.
- Contribution 1: "independence of the protected attributed": missing what it should be independent from.
- I think it's a stretch to describe the C2ST they use as novel. Using Brunner-Munzel instead of some other test for the null that the AUC is 0.5 is, in my opinion, a reasonable but very minor variation.
- The characterization of the C2ST test used by Chakravarti et al. (2023) as relying on an asymptotic normal distribution seems to be slightly inaccurate; Chakravarti et al. mention using three different methods to derive the null distribution: the asymptotic distribution, bootstrap, and permutation methods. It's not clear to me that Brunner-Munzel would outperform this test with the bootstrap or permutation methods. Furthermore, the empirical comparison of Brunner-Munzel to the Mann-Whitney U-based tests looked at power but did not look at Type 1 error rates, so it's somewhat limited.
- On the right side of Figure 3, I'm surprised the authors didn't use the Shapley values for the inspector model $g_\psi$, as in the "Explain Equal Treatment Inspector" stage of their process in Figure 1. That seems like it would have been the natural choice. Currently, that stage is not illustrated in the main paper anywhere.
- Wouldn't it make more sense to call the fairness definition Explanation *Parity*, and reserve "disparity" for violations of that definition?

---

> ### Author Response · Authors · 2025-05-13
>
> Dear reviewer,
>
> Many thanks for your comments.
>
> A general comment is that we understand FTU as a "policy" (an action) on the model, not a metric. The question is which metric measures the results of this policy? We argue the DP does not faithfully measure the results of this policy, but our proposed metric does it better.
>
> We define "better" by collecting a set of requirements based on a use case (explained in Appendix A), and we argue that a good metric for the FTU policy should satisfy them (R-0 to R-5. ). Those criteria are further defined in Appendix A.
>
> In a practical experiment, in Figure 3 (left), it can be seen how DP does not capture the discrimination, where ED does. Furthermore, in Table 6, more experiments are provided. Furthermore, the power test comparisons of the AUC are depicted in Figure 7 and in the corresponding section of the Appendix.
>
> We do not refer to ethical notions but to distributive justice ones. These notions are very important in resource allocation scenarios and are distinct that moral/ethical schools of thought (even if there are connections).
>
> Do please let us know if this high-level explanation helps to understand the scope of the paper. We will be happy to adapt the introduction if the reviewer finds it necessary.
>
> Below are answers to some of the questions posed:
>
> >Why are they using a C2ST to measure violations of demographic parity?
>
> DP is a measure of the independence of the protected attributes on the predictions. It can be measured in many ways; C2STS is one of them. During the paper, for a baseline/ablation study comparison. See Table 6 for a comparison with other ways to measure DP.
>
> >Why is equalised odds introduced if it's never measured?
>
> Where do we introduce equalised odds? We do introduce equal opportunity.
>
> The definition of 2 aims to show that EO is a metric that relies on ground truth data. Then, later in Appendix A3, the metric is discussed.
>
>
> >On the right side of Figure 3, I'm surprised the authors didn't use the Shapley values for the inspector model, as in the "Explain Equal Treatment Inspector" stage of their process in Figure 1. That seems like it would have been the natural choice. Currently, that stage is not illustrated in the main paper anywhere.
>
> Using Shapley values to explain the Explanation Shift Detector is indeed the natural way. We have avoided it for clarity.
>
> In Figure 1, what we have is Explanation Distributions. Shapley values are one type of explanation.

---

> > ### Comment · Reviewer_XmXv · 2025-05-30
> > **Response to authors**
> >
> > Thank you for your response. I agree that FTU can be understood as a policy or modeling approach and that it is not a metric. However, I am unsure what is meant by the question "which metric measures the results of this policy?" What results do you mean? If you just mean the general behavior the model, e.g. the joint distribution of its inputs and outputs, then I'm not sure what this has to do with FTU, since FTU (or the absence thereof) doesn't imply anything about the model in general. Again, I'm also not aware of anywhere in the literature that anyone has argued that demographic parity can be understood as somehow measuring the results of FTU; if anything, as in the Simons paper, researchers have noted that they do not imply each other, which seems at odds with the premise of the paper.
> >
> > I conjecture with reasonable confidence that Explanation Disparity also can be satisfied or not satisfied regardless of whether FTU is or isn't satisfied. If there is some logical relationship between the two of them, then that would be interesting, but it would still remain unclear what the use of Explanation Disparity is. To check whether FTU is satisfied, you simply need to check whether the sensitive feature is included in the model inputs. If the model inputs aren't available, then that presents an interesting problem, but then Explanation Disparity also can't be computed, so it doesn't seem useful in that setting.
> >
> > Overall, even though as I said I find Explanation Disparity to be an interesting metric, I'm afraid I still find the motivation of the paper unclear, and I'm not sure what problem it is supposed to be solving. I think addressing this requires more than just reworking the introduction. Maybe it would make sense to ground ED directly in the relevant distributive justice principles rather than in relation to FTU, but I'm not sure if that would align with the intent of the paper.

---

> > > ### Author Response · Authors · 2025-06-01
> > >
> > > Dear Reviewer,
> > >
> > > Thank you very much for your response.
> > >
> > > At this stage, it remains unclear what the conceptual misunderstanding regarding the scope of the paper might be. In light of the latest discussion—particularly around policy (modelling) vs. metric, notions of distributive justice, and the related metric discussed in Appendix C—we would appreciate it if you could revisit the introduction and indicate which specific notions are found unclear.
> > >
> > > Regarding the second paragraph (concerning the conjecture): if the protected attribute is included in the training data, is it sufficient to assess fairness by analyzing the Shapley values associated with the protected attribute? In Appendix D, we provide a mathematical analysis addressing this point. Please let us know if any part remains unclear or would benefit from further elaboration.
> > >
> > > Lastly, we include below a comment that was mistakenly directed to another reviewer.

---

> > > > ### Author Response · Authors · 2025-06-01
> > > >
> > > > > I think it's a stretch to describe the C2ST they use as novel. Using Brunner-Munzel instead of some other test for the null that the AUC is 0.5 is, in my opinion, a reasonable but very minor variation.
> > > >
> > > > We comment at the end of Section 4.2.1 that the Brunner-Munzel is preferable over other choices: the Wilcoxon–Mann–Whitney test assumes same-shape distribution between positives and negatives; the Fligner–Policello assumes symmetric distributions. We agree that the choice of a specific test is a minor variation, but, among all variations, it is important to choose the one that works best. See also next answer.
> > > >
> > > > > The characterization of the C2ST test used by Chakravarti et al. (2023) as relying on an asymptotic normal distribution seems to be slightly inaccurate; Chakravarti et al. mention using three different methods to derive the null distribution: the asymptotic distribution, bootstrap, and permutation methods. It's not clear to me that Brunner-Munzel would outperform this test with the bootstrap or permutation methods.
> > > >
> > > > Thanks for this comment. Following your question, we implemented the bootstrap and permutation variants. Preliminary experiments show that the power of such variants is comparable to the Brunner-Munzel test, which however is faster (unsurprisingly). Since the statistical test is not a key point (see previous answer), and actually it can be a parameter method of the approach, we will claim that the state-of-the-art method should be used, and that the default method used in our implementation is the Brunner-Munzel test based on both performance and computational efficiency. Appendix G.1 will be expanded reporting experiments with the bootstrap and permutation methods. Also, we will experiment with additional methods we recently were aware of, ed in particular: https://doi.org/10.1002/bimj.201500105. The main objective of Appendix G.1 is to discuss possible variants of the test method.
> > > >
> > > > > Furthermore, the empirical comparison of Brunner-Munzel to the Mann-Whitney U-based tests looked at power but did not look at Type 1 error rates, so it's somewhat limited.
> > > >
> > > > Figure 7 in Appendix G.1 reports power at variation of $\mu$ ranging from 0.02 to 0.50 at significance level $\alpha=0.05$. Notice that for $\mu$ equal to 0.00 the plot would show the Type 1 error. While this is not shown in the figure, the trend of the lines is clear: Type 1 error of Brunner-Munzel is is close to the expected $\alpha=0.05$, while Type 1 error of Wilcoxon–Mann–Whitney approaches $0$ (and the same for the accuracy test). In summary, Wilcoxon–Mann–Whitney is less likely to detect real dependence, which in our context, means it is less likely to detect disparity. However, considering the bootstrap and permutation variants by Chakravarti et al. (2023), their Type 1 error is comparable to the one of the Brunner-Munzel. We will expand the discussion in Appendix G1 to include Type 1 error of the various methods.

---

### Decision · Action_Editor_aFqp · 2025-06-17

**Recommendation:** Reject

**Additional Comments:**

One reviewer recommended rejection while the two other reviewers leaned toward acceptance. I note however that one of the reviewers who leaned toward acceptance did so with reservations.

After reading the reviews and discussions with the authors, as well as the introduction and parts of the paper myself, I find myself in agreement with Reviewer XmXv that the motivations for the paper are unclear. Please see the "claims and evidence" section above for elaboration. I recommend that the authors carefully reconsider these motivations and revise them to be clearer.

In addition, the authors could take this opportunity to address Reviewer XmXv's comments about statistical tests, for example by discussing the results of alternative tests as done in the rebuttal.

**Audience:**

Yes

**Audience Explanation:**

Reviewers generally agree that Explanation Disparity is a new and interesting metric, and thus relevant to the ML fairness community.

**Claims And Evidence:**

No

**Claims Explanation:**

The central claim of the paper is that the proposed Explanation Disparity metric "measures the longstanding criticism of the policy of equal treatment as blindness." This thesis however is rather unclear.
- As Reviewer XmXv notes, "equal treatment as blindness" is operationalized in ML as "fairness through unawareness" (FTU), and assessing the latter amounts to checking whether the model inputs include a protected attribute. The reviewer goes on to say that this would be a challenging research problem in a setting where one does not have access to model inputs, only outputs, but this does not appear to be the setting of the paper.
- It seems therefore that the authors have in mind some broader desired properties, which the model/FTU policy are "criticized" for failing to meet. However, it is not clear what these properties are exactly. Some principles (R0)-(R5) are listed in Section 2.3, but both Reviewer XmXv and I find the discussion of them to be insufficient. Demographic Parity (equality in distributions of predicted outcomes) is criticized as potentially "indicating fairness, even when groups are being treated differently by the model." This suggests that the ideal notion is whether the model exhibits *any* evidence of treating (otherwise similar) individuals from different groups differently, but this is not clear.
- Reviewer XmXv also suggests that grounding Explanation Disparity in principles of distributive justice might be a more direct approach.

If the ideal criterion is indeed whether the model shows any sign of treating individuals differently, then I also think that more attention should be given to the comments of Reviewers rCem and RgW9 on how well explanation algorithms can measure these differences in practice. These comments touch on the reliability of such explanations in general, errors introduced by Shapley value approximation, behavior with alternative explanation algorithms, etc. While the appendices do address some of these points, I think they could be incorporated at least in part into the main paper, and more could be done, to address the question of how well explanations measure differential treatment.

**Resubmission Of Major Revision:**

The authors may consider submitting a major revision at a later time.